# Accurate prediction of CDR-H3 loop structures of antibodies with deep learning

Hedi Chen[1], Xiaoyu Fan[1†], Shuqian Zhu[1†], Yuchan Pei[2], Xiaochun Zhang[1], Xiaonan Zhang[3], Lihang Liu[3], Feng Qian[1*], Boxue Tian[1*]

[1]MOE Key Laboratory of Bioinformatics, State Key Laboratory of Molecular Oncology, School of Pharmaceutical Sciences, Tsinghua University, Beijing, China; [2]Tsinghua Institute of Multidisciplinary Biomedical Research, Tsinghua University, Beijing, China; [3]Department of Natural Language Processing, Baidu International Technology (Shenzhen) Co Ltd, Shenzhen, China

*For correspondence:
qianfeng@tsinghua.edu.cn (FQ);
boxuetian@mail.tsinghua.edu.
cn (BT)

†These authors contributed equally to this work

**Abstract** Accurate prediction of the structurally diverse complementarity determining region heavy chain 3 (CDR-H3) loop structure remains a primary and long-standing challenge for antibody modeling. Here, we present the H3-OPT toolkit for predicting the 3D structures of monoclonal antibodies and nanobodies. H3-OPT combines the strengths of AlphaFold2 with a pre-trained protein language model and provides a 2.24 Å average $RMSD_{C\alpha}$ between predicted and experimentally determined CDR-H3 loops, thus outperforming other current computational methods in our non-redundant high-quality dataset. The model was validated by experimentally solving three structures of anti-VEGF nanobodies predicted by H3-OPT. We examined the potential applications of H3-OPT through analyzing antibody surface properties and antibody–antigen interactions. This structural prediction tool can be used to optimize antibody–antigen binding and engineer therapeutic antibodies with biophysical properties for specialized drug administration route.

## eLife assessment

This article presents H3-OPT, a deep learning method that effectively combines existing techniques for the prediction of antibody structure. This work, supported by **convincing** experiments for validation, is **important** because the method can aid in the design of antibodies, which are key tools in many research and industrial applications.

## Introduction

Antibodies protect the host by recognizing and neutralizing foreign microbes or viruses and provide immunity against future infections. Furthermore, the use of therapeutic antibodies is steadily increasing in clinical treatments for infectious diseases, cancers, and autoimmune disorders (*Meganck and Baric, 2021*; *Kaplon and Reichert, 2021*). For example, anti-PD-L1 antibodies block PD-L1/PD-1 interactions to inhibit tumor growth in patients (*Brahmer et al., 2012*). A typical monoclonal antibody (mAb) is composed of heavy and light chains, while a nanobody (Nb) has only a single-domain variable heavy chain. The CDR heavy chain 3 (CDR-H3) loop is the most variable region in both length and amino acid sequence, and it plays a central role in antigen binding for both mAbs and Nbs. The development of effective therapeutic antibodies requires the solved structures of candidate antibodies, including the CDR-H3. However, it is both costly and labor intensive to experimentally obtain structures, and thus computationally predicted structures are often used to guide antibody design (*Aguilar Rangel et al.,*

*2022*). Despite major advances in computational methods, structure prediction for CDR-H3 loops remains challenging (*Valdés-Tresanco et al., 2023*).

Traditional template-based modeling methods, such as RosettaAntibody (*Adolf-Bryfogle et al., 2018*), PIGS (*Marcatili et al., 2008*), ABodyBuilder (*Leem et al., 2016*), and MODELLER (*Eswar et al., 2006*), often provide demonstrably inaccurate predictions for CDR-H3 sequences (*Almagro et al., 2014*; *Teplyakov et al., 2014*; *Almagro et al., 2011*), and as a result, alternative artificial intelligence (AI)-based approaches, such as AlphaFold2 (AF2) (*Jumper and Hassabis, 2022*; *Jumper et al., 2021*), trRosetta (*Du et al., 2021*), and RoseTTAFold (*Baek et al., 2021*), are increasing in popularity. AF2 has shown comparable accuracy to experimentally determined structures by capturing physical and biological information about protein folding, thus providing a versatile deep learning framework for structure prediction. Structure prediction methods using pre-trained protein language models (PLMs), for example, HelixFold-Single (*Fang et al., 2022*),OmegaFold (*Wu et al., 2022a*), and ESMFold (*Lin et al., 2023*), have shown comparable performance to AF2 with accelerated prediction speed. PLMs can be trained with datasets comprising tens of millions of unlabeled protein sequences in a self-supervised manner and can be subsequently applied to a variety of downstream tasks, such as drug-gable protein target prediction (*Chen et al., 2023*), predicting protein function, and protein design (*Hie et al., 2024*; *Ferruz et al., 2022*; *Madani et al., 2023*). Antibody-specific tools such as IgFold (*Ruffolo et al., 2023*), tFold-Ab (*Wu et al., 2022b*), DeepAb (*Ruffolo et al., 2022*), and NanoNet *Cohen et al., 2022* have also been developed to improve accuracy in CDR-H3 prediction. Among them, IgFold leverages sequence representations from PLMs to efficiently predict antibody structures within seconds, and notably, IgFold can provide accuracy comparable to AF2, enabling high-throughput prediction of antibody structures.

In this study, we present H3-OPT, which combines features of AF2 and PLMs to predict antibody structures. We compare H3-OPT with several other antibody structure prediction methods and found that it can provide a lower average $RMSD_{C\alpha}$ for CDR-H3 loops than other algorithms in three subsets of varying difficulty. To further validate our model, we experimentally solved the structures of three anti-VEGF nanobodies predicted by H3-OPT (*Zhu et al., 2023*). We examined the potential applications of H3-OPT through analyzing antibody surface properties and antibody–antigen interactions. We demonstrate the informative value of high-quality H3 loops for predicting binding affinity and further support the use of H3-OPT as a powerful and versatile tool for studying antigen–antibody interactions. This structural prediction tool can be used to optimize antibody–antigen binding and engineer therapeutic antibodies with biophysical properties.

## Results

### High-quality antibody crystal structures in benchmarks

To evaluate the performance of AF2 against currently available methods, we conducted two datasets (DB1 and DB2) with high-resolution (<2.5 Å) X-ray crystal structures from SAbDab (*Raybould et al., 2020*). The CDR length distributions were similar in each dataset (*Figure 1b*). Additionally, we plotted sequence logos and identified the CDR loops of all heavy chain sequences (*Figure 1c and d*). These sequence logo plots revealed higher degree of sequence variability of CDR loops, particularly in the CDR-H3, with smaller residue letters than framework regions. Overall, high-quality and diverse antibody datasets allowed us to assess the quality of predicted models and evaluate the strengths and weaknesses of all alternative methods.

### Accuracy of AF2 for the overall antibody structure predictions was remarkable

To assess the similarity between the predicted models generated by each method and the experimentally determined structure, we first computed the template modeling scores (TM-scores) (*Zhang and Skolnick, 2004*) for all predicted models of each target. The TM-scores of AF2 predictions in different datasets were as follows: DB1, 0.93 ± 0.04; DB2, 0.94 ± 0.03 (*Figure 1e*). Although the average TM-score of AF2 was slightly lower than those of DeepAb in DB1, it outperformed DeepAb by a large margin in DB2 (0.94 vs 0.87, on average). Then, we used global distance test (GDT) scores (*Zemla et al., 1999*) to assess the similarity of predicted substructures at different structural cutoffs. The average GDT-TS scores of the models generated by AF2, ABodyBuilder, and DeepAb in DB1 were

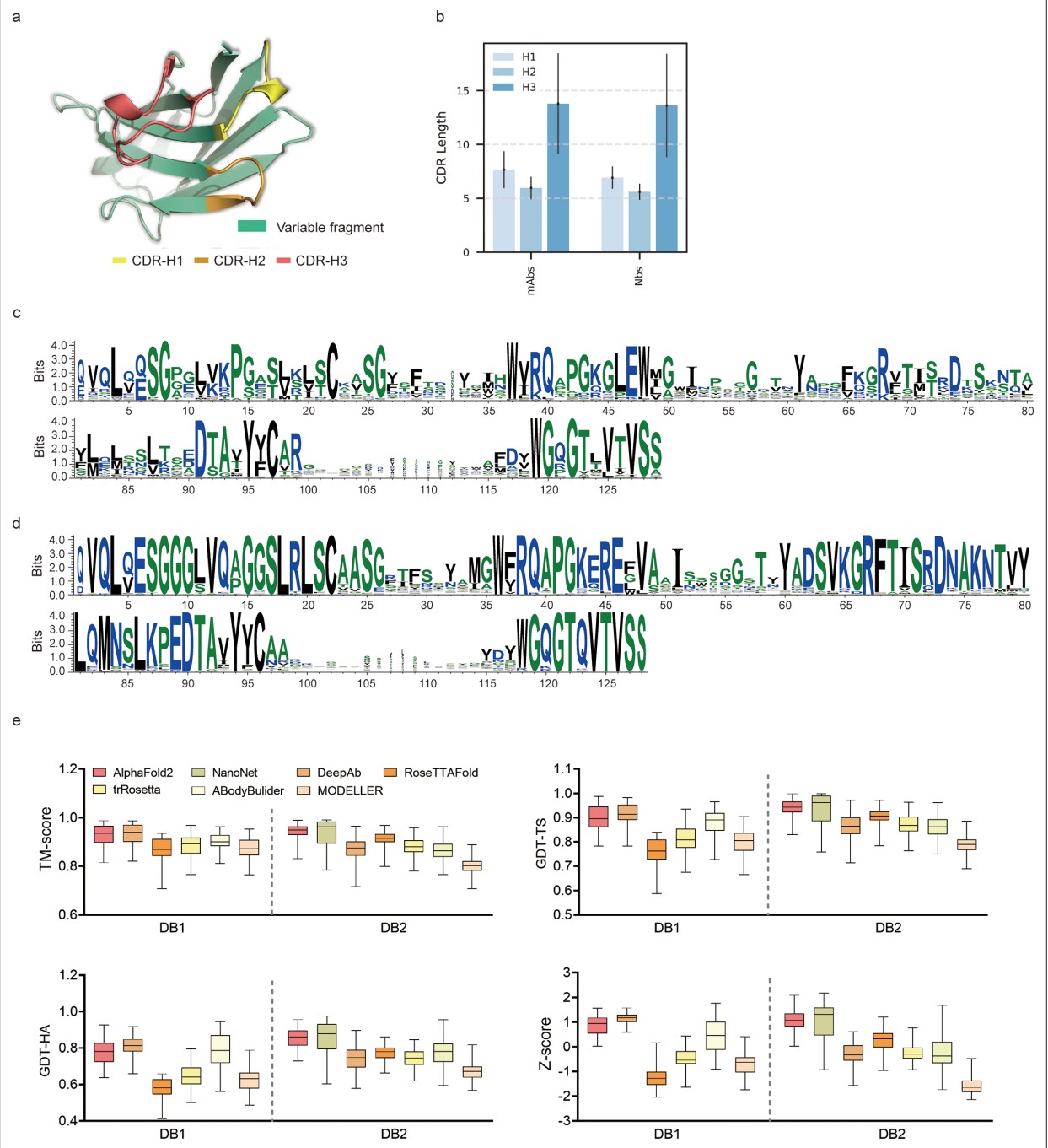

**Figure 1.** Accuracy of AF2 on antibody modeling. (**a**) Schematic for CDR heavy chain loops. (**b**) The CDR lengths of monoclonal antibodies (mAbs) (n = 47) and nanobodies (Nbs) (n = 78). The error bars represent the standard deviation of the data. (**c**) Sequence logo plots of $V_H$ fragments in DB1. (**d**) Sequence logo plots of $V_H$ fragments in DB2. The different colors of codes represent the hydrophobicity of amino acids. (**e**) The performance of AF2 on different datasets using various evaluation metrics. In the box plots, the lines at the center of the boxes represent the medians, and the top and bottom lines of the boxes represent the upper and lower quartiles.

0.90, 0.88, and 0.91, respectively. In DB2, the GDT-TS scores of AF2, RoseTTAFold, and NanoNet all exceeded 0.90. Furthermore, we found that the models with high GDT-TS scores also exhibited higher GDT-HA scores compared to the models with low GDT-TS scores. We finally calculated the average Z-scores of GDT-TS, GDT-HA, and TM-score for all datasets to provide comprehensive scores for each method. AF2 outperformed other methods on both DB1 and DB2, with an average Z-score of 0.87 and 1.08, respectively.

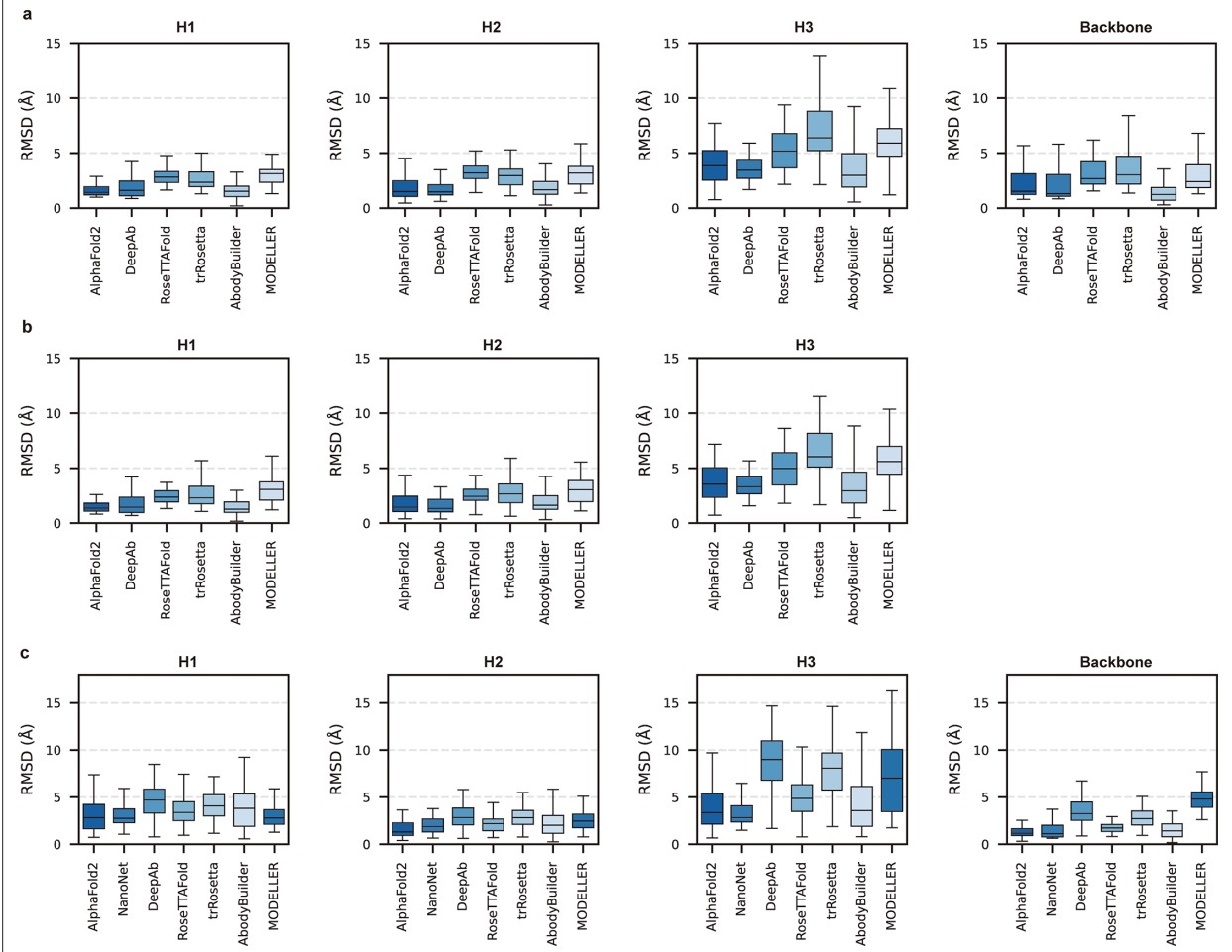

**Figure 2.** Accuracy of AF2 on different antibody regions. (**a**) The performance of AlphaFold2 in DB1 relative to other methods after superimposing Fv backbones. (**b**) The performance of H3-OPT in DB1 relative to other methods after superimposing $V_H$ backbones. (**c**) The performance of H3-OPT in DB2 relative to other methods.

The online version of this article includes the following figure supplement(s) for figure 2:

**Figure supplement 1.** Local accuracy of AlphaFold2 prediction.

## AlphaFold2 accurately predicted the $V_H$/$V_L$ orientations and CDR-H3 loops of antibodies

Given the vital roles of CDR-H3 loops in antigen recognition, we assessed the local accuracy of CDR-H3 loops among all participants. We first superimposed the backbone atoms of the entire Fv regions to reference structures and calculated the heavy atom RMSDs of CDR-H3 loops ($RMSD_{HA}$). The average $RMSD_{HA}$ values of AF2, DeepAb, and ABodyBuilder in DB1 were 3.92, 3.64, and 3.69 Å, respectively, which were lower than those of other methods (*Figure 2a*). Additionally, these $RMSD_{HA}$ values decreased slightly after superimposing the $V_H$ backbone heavy atoms, indicating that these methods could predict $V_H$/$V_L$ orientations more accurately than other alternative methods (*Figure 2b*). Similar to TM-score, DeepAb performed better than AF2 in DB1, but its accuracy decreased substantially in DB2 (*Figure 2c*). AF2, NanoNet, and ABodyBuilder outperformed other methods in DB2, with average $RMSD_{HA}$ values of 3.79, 3.44, and 4.37 Å, respectively.

To better understand the prediction results, we next conducted side-to-side comparisons for the abovementioned methods. The results showed that there were no significant differences in either the backbone or CDR-H3 RMSDs after superimposing the $V_H$ backbone heavy atoms between DeepAb and AF2 (*Figure 2—figure supplement 1a*). Additionally, AF2 accurately produced the Nb structures with lower backbone RMSDs (all below 4 Å), but provided comparative accuracy in CDR-H3 RMSDs

**Table 1.** The root mean square deviation (RMSD) results of PM6D3 level re-ranking method on 14 same CDR-H3 antibodies.

| PDB ID | Ranked 0 RMSD | Lowest energy RMSD | Lowest RMSD | Δ$_{RMSD}$* |
|---|---|---|---|---|
| 4kmt | 1.06 | 1.14 | 1.05 | –0.08 |
| 5i19 | 2.16 | 1.91 | 1.77 | 0.25 |
| 5i1l | 3.80 | 3.20 | 3.19 | 0.60 |
| 5i17 | 2.86 | 3.71 | 2.86 | –0.85 |
| 5i1d | 2.10 | 2.10 | 2.02 | 0.00 |
| 5i1c | 2.43 | 1.66 | 1.45 | 0.77 |
| 5i1a | 2.37 | 0.85 | 0.59 | 1.52 |
| 5i1i | 3.72 | 3.51 | 3.51 | 0.21 |
| 5i15 | 2.16 | 1.94 | 1.35 | 0.22 |
| 5i16 | 3.19 | 1.70 | 1.39 | 1.49 |
| 5i18 | 2.88 | 2.88 | 2.88 | 0.00 |
| 5i1e | 1.62 | 1.13 | 0.92 | 0.49 |
| 5i1g | 2.08 | 2.08 | 2.00 | 0.00 |
| 5i1h | 1.58 | 1.84 | 1.32 | –0.26 |

*Δ$_{RMSD}$ was calculated by subtracting the RMSD of predicted model from the RMSD of Ranked_0 model.

with NanoNet (**Figure 2—figure supplement 1b**). We found that the main reason for this result was that NanoNet predicted wrong main-chain structures at the C-terminus. ABodyBuilder utilized homology modeling-based algorithm for structure prediction, providing similar or better backbone quality than AF2 (**Figure 2—figure supplement 1c and d**). However, it did not significantly improve the accuracy of the CDR3 loops because ABodyBuilder highly relied on the quality of templates (especially when modeling DB2). Taken together, these results indicated that AF2 was a highly effective tool for predicting antibody structures both in mAbs and Nbs, and it produced CDR-H3 loops that were comparable to those generated by AI-based antibody-specific methods.

## Quantum mechanics-based optimization failed to optimize CDR-H3 loops

Due to the lack of consideration of electronic effects in AF2, we hypothesized that the accuracy of CDR-H3 could be further improved by quantum mechanics (QM)-based methods. We applied two QM-based approaches to optimize AF2 models: the first was an energy-based re-ranking method, and the second was loop optimization with QM. For the first approach, we introduced a new dataset

**Table 2.** Accuracy of quantum mechanics (QM)-based re-ranking methods.

| Method | Freeze terminal Cα | CDR* | Phase | Ranked 0 RMSD | Lowest energy RMSD | Lowest RMSD | Δ$_{RMSD}$ |
|---|---|---|---|---|---|---|---|
| PM6D3 | Y | H3 | Gas | 2.64 | 2.76 | 2.16 | –0.12 |
| PM6D3 | N | H3 | Gas | 2.53 | 2.67 | 2.03 | –0.14 |
| PM6D3 | N | H1, H2, H3 | Gas | 2.50 | 2.64 | 2.00 | –0.14 |
| B3LYP | N | H3 | Gas | 2.66 | 2.87 | 2.30 | –0.21 |
| B3LYP | N | H3 | Water | 2.66 | 2.68 | 2.30 | –0.02 |

RMSD = root mean square deviation.

*CDR means the energy of which loop is used to re-rank AF2 models.

**Table 3.** Accuracy of quantum mechanics (QM)-based optimization methods.

| Method | Freeze terminal Cα | Structure generation method | Phase | Ranked 0 RMSD | Lowest energy RMSD/opted RMSD | Lowest RMSD | $\Delta_{RMSD}$ |
|---|---|---|---|---|---|---|---|
| PM6D3 | Y | / | Gas | 1.69 | 1.74/1.87 | 1.37 | –0.05 |
| B3LYP | N | / | Gas | 1.63 | 1.65/2.55 | 1.38 | –0.02 |
| B3LYP | N | / | Water | 1.63 | 1.58/2.25 | 1.38 | 0.05 |
| B3LYP | N | Boltzmann | Gas | 1.56 | 2.05 | 1.28 | –0.49 |
| B3LYP | N | Boltzmann | Water | 1.56 | 1.81 | 1.28 | –0.25 |
| B3LYP | N | Boltzmann, minimized | Gas | 1.56 | 1.96 | 1.28 | –0.40 |
| B3LYP | N | Boltzmann, minimized | Water | 1.56 | 1.84 | 1.28 | –0.28 |

RMSD = root mean square deviation.

containing 14 antibody sequences with the same CDR-H3 and then run QM optimization for all AF2 models. These models were re-ranked according to the QM energies. As shown in *Table 1*, our results demonstrated that this method outperformed the default ranking criteria for 8 out of 14 targets, achieving an average RMSD improvement of 0.69 Å. However, this re-ranking method did not yield more accurate results when applied to a larger database (DB3) (*Table 2*). We also made additional attempts to improve the re-ranking process by removing constraints on terminal atoms, altering re-ranking criteria, and considering solvent effects. However, these methods did not improve the accuracy of AF2 (ΔRMSDs < 0). Moreover, the QM-based optimization methods also failed to improve the accuracy of H3 loops (*Table 3*). We finally generated structures based on the conformational proportions using Boltzmann distribution of the QM energies, but the accuracy of these structures did not match that of Ranked_0, with all ΔRMSD below 0. In conclusion, although QM-based methods may not improve the accuracy of CDR-H3 loops in general cases, it still provided opportunities for loop modeling as the development of more accurate physics-based methods.

## Molecular dynamics (MD) simulations could not provide accurate CDR-H3 loop conformations

MD simulations were extensively used to explore stable conformations of proteins in a water environment (*Pan and Aller, 2018*) and were also introduced to loop modeling (*Barozet et al., 2021*). We used MD simulations to search for representative CDR-H3 loop conformations. The simulation

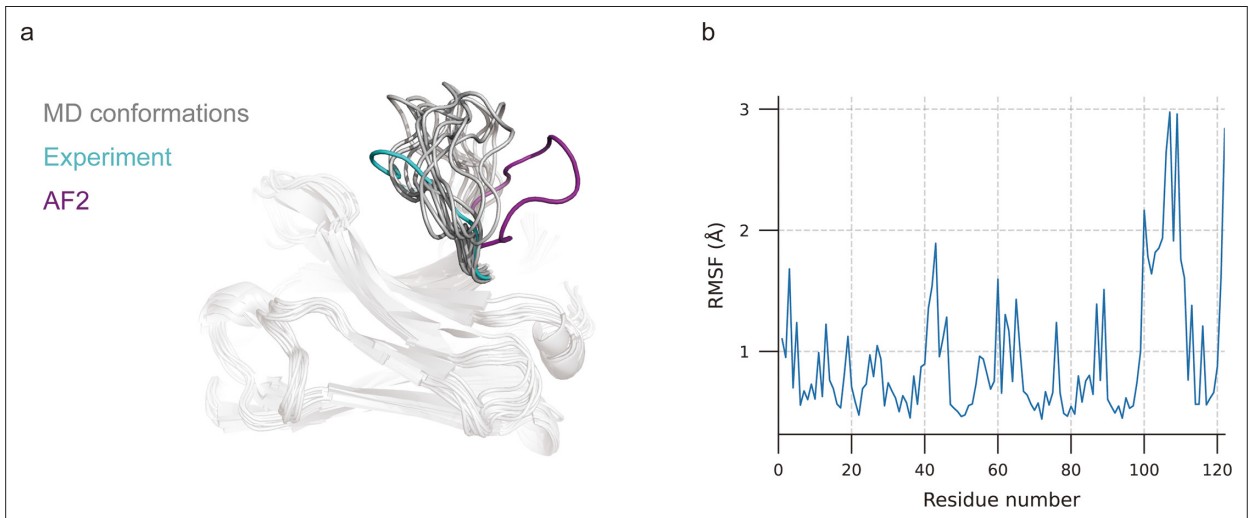

**Figure 3.** Molecular dynamics (MD) generated conformations for benchmark target 7N0R. (**a**) Comparison of CDR-H3 loops of MD (gray), AF2 (pink), and experimentally determined structure (cyan). (**b**) Root mean square fluctuation (RMSF) of antibody residues during simulation. CDR-H3 loop is located in residue number ranging from 98 to 109.

**Table 4.** The accuracy of molecular dynamics (MD)-based CDR-H3 loop optimization in the 10 worst cases of AF2.

| PDB ID | Cα-RMSD$_{Ranked\_0}$ | Cα-RMSD$_{MD\_opt}$ | ΔCα-RMSD |
|--------|----------------------|---------------------|----------|
| 7n0r | 10.92 | 5.62 ± 0.97 | 5.30 |
| 3juy | 6.37 | 5.71 ± 0.23 | 0.66 |
| 5y80 | 6.61 | 7.59 ± 0.47 | –0.98 |
| 7a4t | 6.19 | 7.48 ± 0.29 | –1.29 |
| 4nzr | 6.57 | 7.73 ± 0.26 | –1.16 |
| 6xzu | 7.45 | 6.34 ± 0.94 | 1.11 |
| 6x05 | 6.32 | 7.48 ± 0.63 | –1.16 |
| 3c08 | 6.68 | 7.01 ± 0.11 | –0.33 |
| 4z9k | 9.04 | 8.01 ± 0.37 | 1.03 |
| 6oca | 7.61 | 8.01 ± 0.34 | –0.40 |

RMSD = root mean square deviation.

systems consist of antibody structures, water, and ions. We selected the top 10 challenging-to-predict targets from the DB1 and DB2 to investigate the dynamic conformations of CDR-H3 loops. The first target was 7N0R, a single-domain antibody that binds the SARS-CoV-2 Nucleocapsid protein with a 12-residue CDR-H3 loop (*Ye et al., 2021*). We found that MD successfully generated conformations with lower CDR-H3 Cα-RMSDs (average value is 5.62 Å, achieving an improvement of 5.30 Å over the AF2 prediction) by correcting the orientation of the CDR-H3 loop (*Figure 3a*). Additionally, the conformations of 7N0R indicated that the CDR-H3 loop was more flexible than framework regions and other CDR loops, demonstrated by root mean square fluctuation (RMSF) of every residue during simulation (*Figure 3b*). Despite these promising results for 7N0R, the high CDR-H3 Cα-RMSDs of all 10 targets (>5 Å) and poor performance in other cases (ΔCα-RMSD < 1.5 Å) (*Table 4*) both suggested that the conformation search method based on MD simulations failed to generate CDR-H3 loops that closely matched native structures. However, the accuracy of these antibodies with relative long CDR-H3 loops could be improved by H3-OPT through incorporating latent structural information of loop folding from ESM2 or enhanced sampling techniques (*Feng et al., 2021*).

## The H3-OPT workflow

Prior to training H3-OPT, we first conducted a thorough evaluation of currently available antibody structure prediction tools, including AF2, RoseTTAFold, trRosetta, NanoNet, DeepAb, MODELLER, and ABodyBuilder. AF2 provided high-accuracy predictions for the overall structures of both mAbs and Nbs, with TM-scores and GDT scores >0.9. Based on the relatively higher accuracy of AF2 in structural predictions, its prediction was used to extract structural features of CDR-H3 loops for further optimization by H3-OPT. Using a mAb/Nb sequence and its Fv model structure from AF2 as input, H3-OPT was then used to generate a refined structure. As data quality has large effects on prediction accuracy, we constructed a non-redundant dataset (sequence identity < 0.8) with 1286 high-resolution (<2.5 Å) antibody structures from SAbDab (*Dunbar et al., 2014*; *Figure 4a*). The dataset was then divided into training, validation, and test sets based on amino acid sequence identity, which was done by using the UCLUST (*Edgar, 2010*) software. To assess the prediction results, the test set was split into three subgroups according to the differences in AF2 accuracy (RMSD, a measure of the difference between the predicted structure and an experimental or reference structure) stemming from the length of CDR-H3 sequence: easy-to-predict targets (0–2 Å, Sub1); moderate-difficulty targets (2–4 Å, Sub2); and challenging-to-predict targets (>4 Å, Sub3), with average CDR-H3 loop sequence lengths of 9.12, 11.08, and 16.43, respectively.

The workflow of H3-OPT is depicted in *Figure 4b*. H3-OPT consists of a template module and a PLM-based structure prediction module (PSPM). The template module determines whether to use PSPM to optimize CDR-H3 and comprises two submodules: a confidence-based module (CBM) and

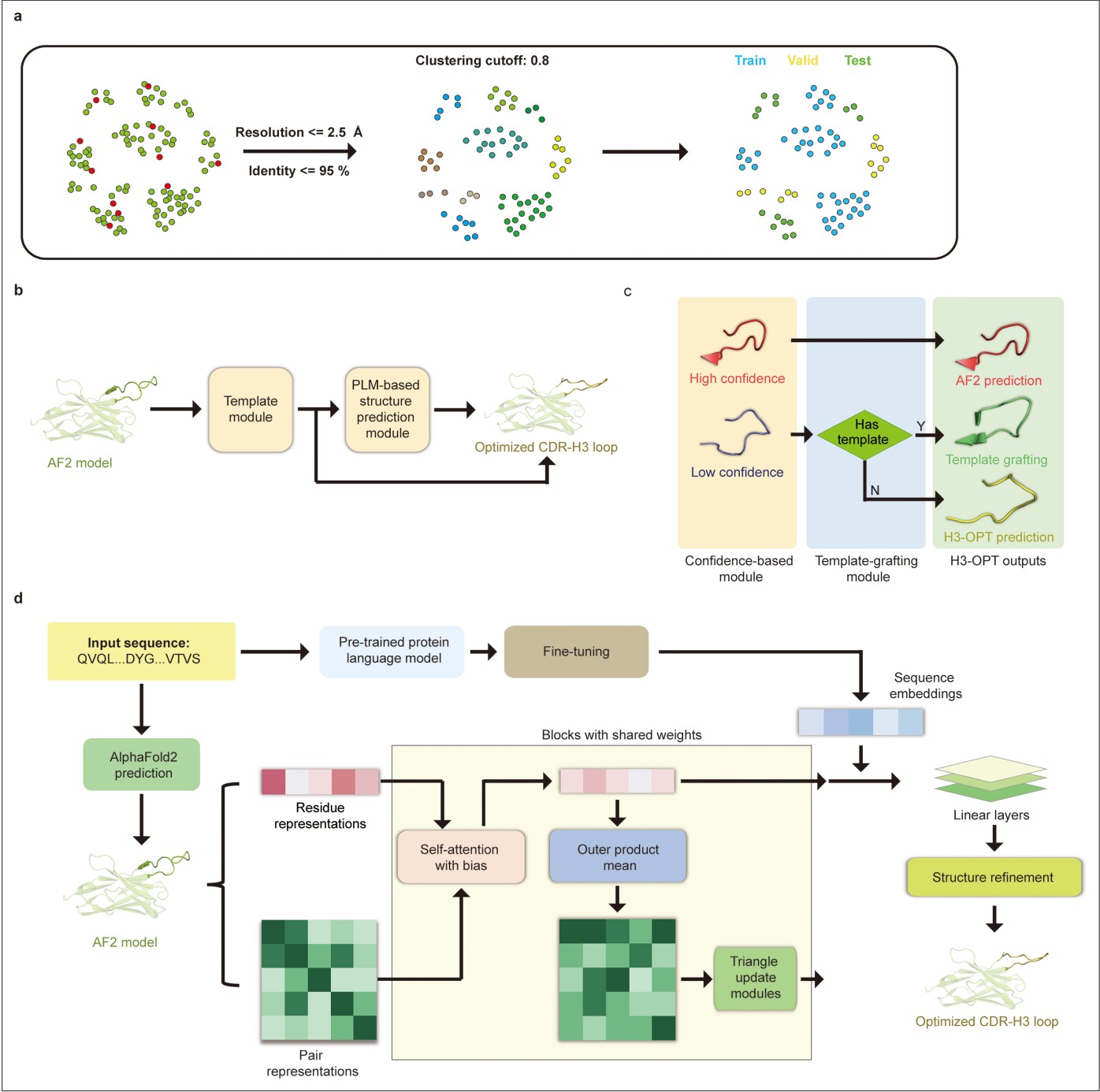

**Figure 4.** H3-OPT architecture. (**a**) Schematic for dataset preparation. Structures were screened from the SAbDab database based on resolution and sequence identity. Clustering of the filtered, high-resolution structures yielded three datasets for training (n = 1021), validation (n = 134), and testing (n = 131). (**b**) The workflow of H3-OPT includes two modules. The template module determines whether to use PLM-based structure prediction module (PSPM), while the PSPM module optimizes the AF2 input structures. (**c**) The template module retains AF2-predicted loops when the confidence score is >0.8 and grafts CDR-H3 loops onto AF2 models for structures with an available template. (**d**) In the PSPM, the network extracts residue-level information and pairwise residue representations from the AF2-predicted models, which are subsequently updated using weight-sharing blocks and concatenated with sequence representations from ESM2. The resulting data is used to predict the 3D coordinates of the H3 loops.

a template-grafting module (TGM). The CBM calculates an AF2 confidence score to evaluate the reliability of CDR-H3 loop sequence inputs through MSA and template searching. The TGM then identifies a template from the H3 template database and grafts the CDR-H3 loop onto the AF2 model if such a template is available (*Figure 4c*).

The PSPM contains a series of attention-based components to ensure that the complete structural context is extracted from the input sequences. Specifically, the PSPM employs a row-wise gated attention layer that updates residue-level information and exchanges information within residue pair representations and residue features to infer relationships between the spatial and residue representations. Additionally, by leveraging sequence-level representations from PLMs along with residue-level information, the PSPM can integrate high-information features, such as protein folding patterns, across a vast protein space to predict final three-dimensional (3D) atomic coordinates (*Figure 4d*).

To better understand the contribution of each module to H3-OPT accuracy with varying factors, we next examined how incorporating different approaches through the template and PSPM modules could improve accuracy of H3-OPT. While AF2 showed markedly stronger predictions than IgFold in Sub1 (*Figure 5a*), IgFold provided higher accuracy in Sub3 (*Figure 6a*). In light of previous studies that showed PLM-based models are computationally more efficient but have unsatisfactory accuracy when high-resolution templates and MSA are available (*Lin et al., 2023*), we therefore sought to combine the strengths of AF2 with PLM-based modeling. Similar to IgFold, an initial version of H3-OPT lacking the CBM could not replicate the accuracy of AF2 in Sub1. Interestingly, we observed that the AF2 confidence score of CDR-H3 shared a strong negative correlation with Cα-RMSDs (Pearson correlation coefficient = −0.67; *Figure 5b*), which led to us to hypothesize that AF2 models with high-confidence scores might be sufficiently accurate, and therefore do not require further optimization. Following this notion, we incorporated a CBM submodule that could directly retain high-confidence structures from AF2. TGM submodule was added as a means of grafting identical H3 loop templates from public PDBs onto AF2-predicted models to further improve accuracy. Ablation studies in which the CBM or TGM were excluded to determine their respective contributions to the final predictive accuracy (*Figure 5c–f*) revealed that H3-OPT without the template module could generate structures with average Cα-RMSDs of 1.76, 2.76, and 4.86 Å for the Sub1, Sub2, and Sub3 datasets, respectively. Incorporating only the CBM into our model significantly improved $RMSD_{C\alpha}$ score by 0.62 Å for Sub1, but exerted negligible effects on Sub2 and Sub3. By contrast, including the TGM alone resulted in substantially improved $RMSD_{C\alpha}$ for Sub2 and Sub3 by 0.68 and 1.04 Å, respectively, suggesting that templates were more effective for relatively long CDR-H3 antibodies. These results suggested that the combination of TGM and CBM modules could leverage available templates to improve prediction accuracy.

Since the large majority of sequences in Sub2 and Sub3 have long CDR-H3 loops with few sequence homologs, attaining high accuracy in structural predictions becomes increasingly challenging for AF2 (*Jumper et al., 2021*). Inspired by IgFold and other PLM-based methods (*Figure 3a*), we thus developed a PSPM module to capture structural information from the sequence embeddings of PLMs (*Rives et al., 2021*). The key innovations of the PSPM for our workflow were the integration of sequence-level representations from PLMs and the simplified architecture of AF2. ProtTrans-T5, AntiBERTy, and ESM2 were initially used without fine-tuning, resulting in overall average Cα-RMSDs of 2.40, 2.49, and 2.32 Å, respectively (*Table 5*). To improve accuracy, we employed a fine-tuning approach for the downstream CDR-H3 structure prediction task. As ESM2 outperformed other PLMs in our test set, we fine-tuned parameters of all ESM2 hidden layers, which resulted in an overall $RMSD_{C\alpha}$ of 2.24 Å for H3-OPT. It should be noted that most computational models, such as IgFold, froze PLM weights during model training (*Ruffolo et al., 2023*). Analysis of test subsets showed that H3-OPT could provide high accuracy in side chain predictions, high TM-scores, and high GDT scores in Sub2 and Sub3 (*Figure 6b and c*).

To expand its generalizability, we simplified the Evoformer architecture and introduced parameter-sharing to directly predict Cartesian coordinates of CDR-H3 loops through both residue-level features and pair representations. In residue-level representations, rows represented amino acid types and structural features derived from AF2, while columns represented individual residues from the input sequence. The pair representations contained information about the residue pairs, such as Cα-Cα distance. These residue-level representations were updated by the attention-based layers and continuously communicated with pair representations; the pair representations were then updated via triangular multiplicative layers (*Jumper et al., 2021*). This simplified architecture improved model efficiency and avoided overfitting. We also applied a structural alignment strategy for feature extraction and model prediction. Given the high conservation of Fv structures, the alignment strategy was designed to effectively capture residue contribution to loop folding, resulting in improved training speed and

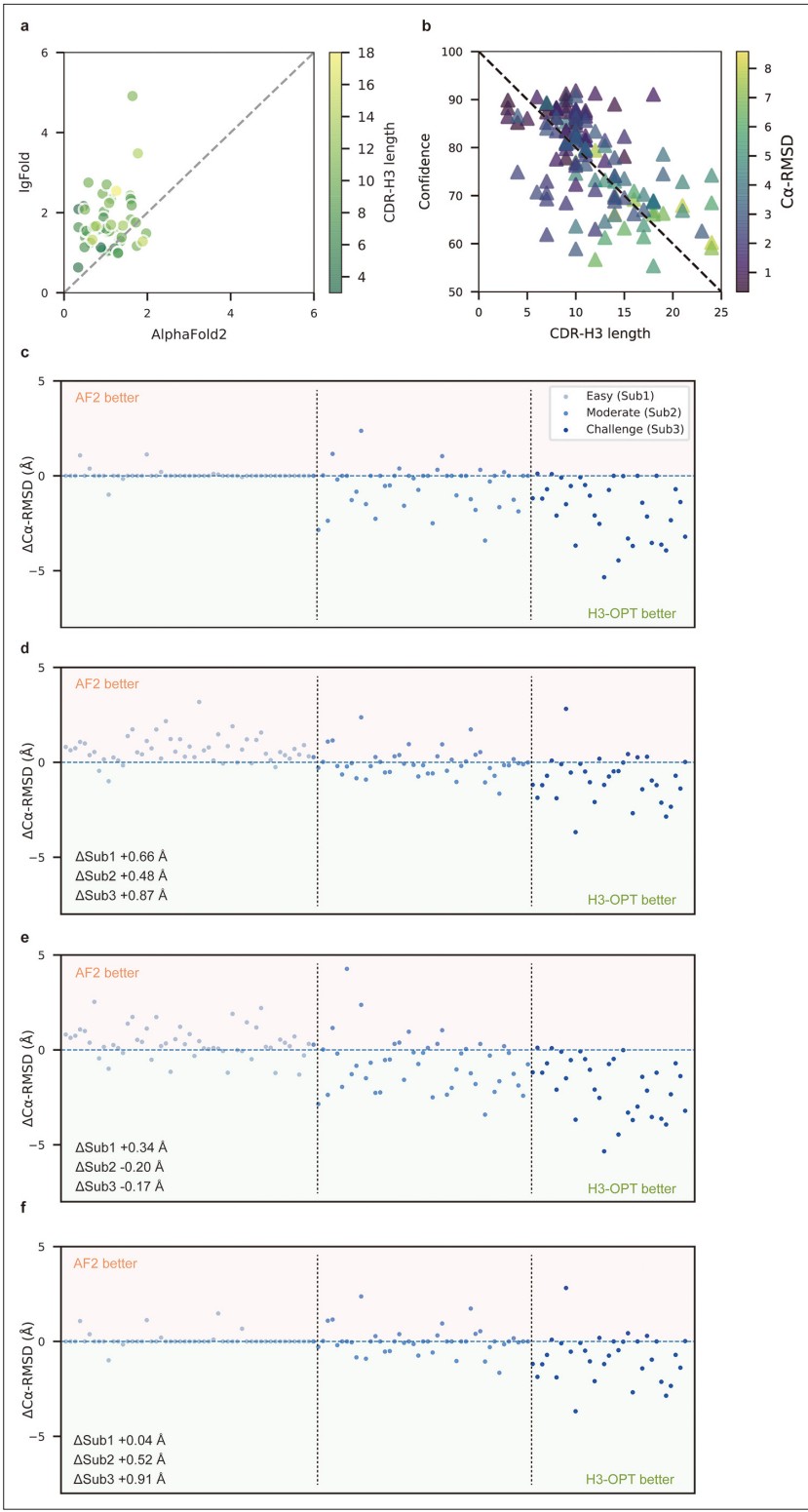

**Figure 5.** Template module and ablation studies. (**a**) Side-by-side comparison of Cα-RMSDs of AF2 and IgFold for Sub1 (n = 52); color scale for data points reflects CDR3 length. AF2 outperformed IgFold for targets left of the dashed diagonal; IgFold outperformed AF2 for targets right of the dashed diagonal. (**b**) Correlations between AF2 confidence score and amino acid sequence length of CDR-H3 loops. Datapoint color indicates Cα-RMSD value for that target. The correlation coefficient for confidence score and CDR-H3 loop length is −0.5921. (**c**) The accuracy of H3-OPT in three subgroups of the test set. ΔCα-RMSDs were calculated by subtracting the RMSD_Cα of AF2

*Figure 5 continued on next page*

*Figure 5 continued*

from that of H3-OPT. AF2 had higher accuracy for targets above the dashed line; H3-OPT had better accuracy for structures below the dashed line. (**d**) Differences in H3-OPT accuracy without the template module. This ablation study means only PLM-based structure prediction module (PSPM) is used. (**e**) Differences in H3-OPT accuracy without the confidence-based module (CBM). This ablation study means input loop is optimized by template-grafting module (TGM) and PSPM. There are 30 targets in our database with identical CDR-H3 templates. (**f**) Differences in H3-OPT accuracy without the TGM. This ablation study means input loop is optimized by CBM and PSPM.

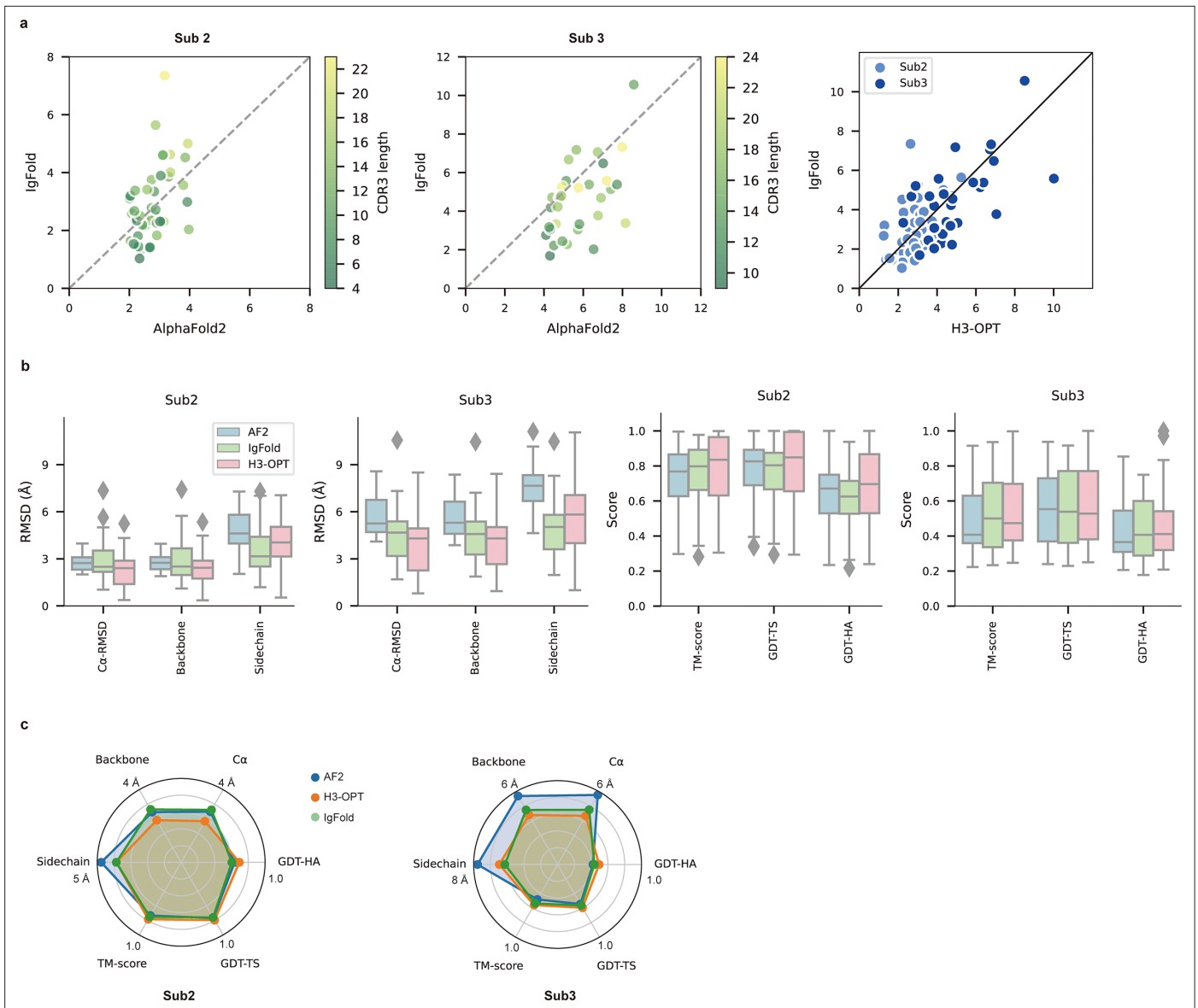

**Figure 6.** PLM-based structure prediction module (PSPM) module. (**a**) Side-by-side comparison of Cα-RMSDs for AF2 and IgFold, IgFold and H3-OPT in the Sub2 (n = 46) and Sub3 (n = 33) test sets, respectively. (**b**) Comparison of prediction accuracy between AF2 and H3-OPT for Sub2 and Sub3 targets. Metrics including root mean square deviations (RMSDs), template modeling scores (TM-scores), and global distance test (GDT) scores were used to quantitatively assess similarity between predicted and experimental structures. (**c**) Comparison of prediction accuracy between AF2 and H3-OPT using six metrics (RMSD$_{Cα}$, RMSD$_{backbone}$, RMSD$_{sidechain}$, TM-score, GDT-TS score and GDT-HA score). Radar plots of the mean values of different methods and metrics in predictions of Sub2 and Sub3 targets.

**Table 5.** Performance of H3-OPT with different protein language models (PLMs).

| | $\text{RMSD}_{C_\alpha}$ (Å) |
| --- | --- |
| H3-OPT | 2.24 ± 1.05 |
| AF2 | 2.85 ± 0.69 |
| ESM2 | 2.31 ± 1.13 |
| Without PLM | 2.41 ± 1.26 |
| AntiBERTy | 2.49 ± 1.42 |
| ProtTrans-T5 | 2.40 ± 1.28 |

RMSD = root mean square deviation.

accuracy. Taken together, H3-OPT provided high-accuracy predictions for CDR-H3 loops, facilitating the development of antibody therapeutics.

## H3-OPT provides higher accuracy CDR-H3 models than existing structure prediction software

We then evaluated the performance of H3-OPT against currently available methods, including AF2, IgFold, HelixFold-Single, ESMFold, and OmegaFold. H3-OPT achieved an average Cα root mean square deviation ($\text{RMSD}_{C_\alpha}$) of 2.24 Å for CDR-H3, while AF2 and IgFold had CDR-H3 RMSDs of 2.85 and 2.87 Å, respectively (*Figure 7a*). HelixFold-Single, OmegaFold, and ESM-Fold generated comparatively poor predictions, with RMSDs for CDR-H3 of 3.39, 3.75, and 4.23 Å, respectively. As shown in *Figure 7b*, H3-OPT provided higher accuracy predictions than other methods in all three subgroups, with average RMSDs of 1.10, 2.28, and 3.99 Å, respectively. While the predictions of H3-OPT were comparable to AF2 in Sub1, its accuracy was higher than AF2 in Sub2 and Sub3.

In light of these results, we then sought to validate H3-OPT using three experimentally determined structures of anti-VEGF nanobodies, including a wild-type (WT) and two mutant (Mut1 and Mut2) structures, that were recently deposited in Protein Data Bank (*Figure 7c*). Although Mut1 (E45A) and Mut2 (Q14N) shared the same CDR-H3 sequences as WT ($\text{Length}_{\text{CDR-H3}}$ = 17), only minor variations were observed in the CDR-H3. H3-OPT generated accurate predictions with Cα-RMSDs of 1.510, 1.541, and 1.411 Å for the WT, Mut1, and Mut2, respectively (the confidence scores of these AlphaFold2-predicted loops were all higher than 0.8, and these loops were accepted as the outputs of H3-OPT by CBM). Subsequent comparison with IgFold showed that AlphaFold2 outperformed IgFold on these targets and IgFold could not accurately predict the short helix in CDR-H3, resulting in Cα-RMSDs of 2.776 Å (WT), 2.888 Å (Mut1), and 2.448 Å (Mut2) and more diverse conformations of Mut1 and Mut2. These results indicated that IgFold was capable of learning long-range correlations in protein sequences, but that these long-range correlations could introduce larger errors when MSA and templates were available. These results thus demonstrated that MSA could strongly influence the accuracy for which similar CDR-H3 template structures were available.

## H3-OPT can predict antibody surface properties

To demonstrate how improved CDR-H3 structural accuracy can assist antibody engineering, we applied H3-OPT to predict surface properties. A comparison of H3-OPT with AF2 for identifying surface amino acids (SAAs) by the relative accessible surface areas (rASAs) of CDR-H3 loops showed that H3-OPT could predict SAAs, on average, close to that of the native structures (9.46 vs 9.40, *Figure 8a*), whereas AF2 predicted an average of 9.81 SAAs. Furthermore, H3-OPT predicted closer values to native structures than AF2 in predicting diverse surface properties, such as the distribution of hydrophilic or charged SAAs (*Figure 8b*).

To examine insights H3-OPT could provide into the biophysical properties of antibodies (*Cong et al., 2021*), we next estimated the solvent-accessible surface areas (SASAs). The average SASAs of H3-OPT loops closely resembled that of the native structures, with larger differences in AF2 predictions (*Figure 8c*). Plots of SASA distributions at each alignment position revealed smaller errors in H3-OPT than AF2, with ΔSASA ranging from −12.17 to 9.74 Å$^2$ (*Figure 8—figure supplement 1*). In

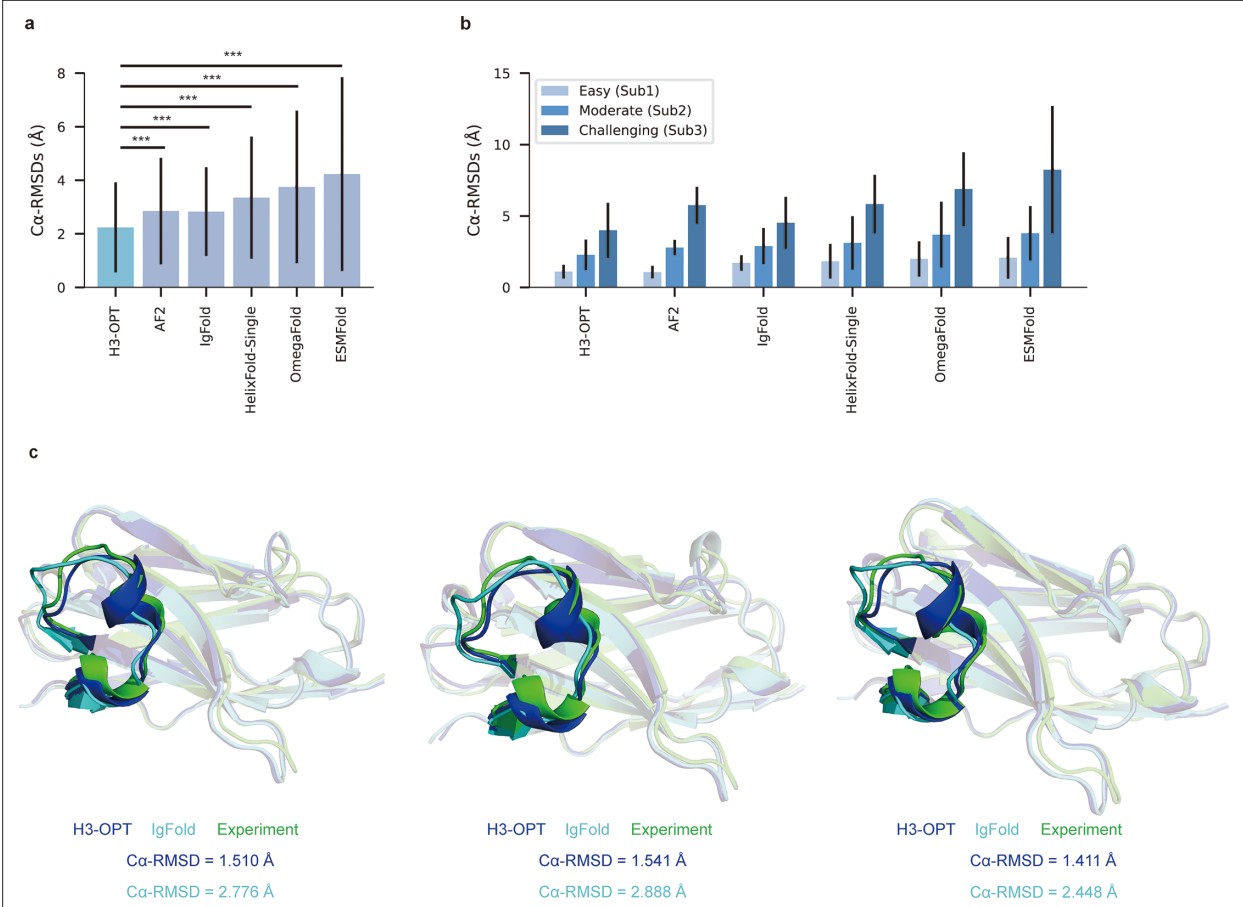

**Figure 7.** Accuracy of CDR-H3 loop prediction by H3-OPT. (**a**) The performance of H3-OPT in the test set ($n_{mAbs}$ = 119, $n_{Nbs}$ = 12) relative to other methods. The $RMSD_{C\alpha}$ of H3-OPT was significantly lower than other existing methods (p<0.001). (**b**) The performance of H3-OPT in structural predictions of three subgroups of the test set (n = 52, 46, and 33). (**c**) H3-OPT structural predictions for three anti-VEGF nanobodies (PDB ID: 8IIU, 8IJZ, 8IJS). The sequence identities of the VH domain and H3 loop are 0.816 and 0.647, respectively, compared with the best template. ***p<0.001. RMSD, root mean square deviation.

The online version of this article includes the following figure supplement(s) for figure 7:

**Figure supplement 1.** Comparison of accuracy between AF2, H3-OPT, and tFold-Ab methods using the CAMEO 2022 benchmark dataset (*Leemann et al., 2023*).

addition, comparison of surface charge distributions on the predicted CDR-H3 loops generated by H3-OPT and AF2 in an electrostatic map of a representative structure (PDB 5U3P, *Figure 8d*) showed that H3-OPT predictions were consistent with the experimental structure. These results collectively showed that accurate surface properties predicted by H3-OPT could provide insights into antibody folding and stability, as well as their interactions with antigens.

## H3-OPT can facilitate investigation of antibody–antigen interactions

To explore the potential applications of H3-OPT in studying antibody–antigen interactions, we analyzed contact patterns at predicted binding sites. Given that predicting the structure of an antibody–antigen complex remains challenging, we superimposed $V_H$ fragments from H3-OPT and AF2 onto experimentally determined native complex structures available in our test set. After identifying the contact residues of antigens by H3-OPT, we found that H3-OPT could substantially outperform AF2 (*Figure 9a*), with a median precision of 0.82 and accuracy of 0.98 compared to 0.71 precision and 0.97 accuracy of AF2. Next, we estimated the distances among interface residues, which are related to binding affinity at the interface. We found that H3-OPT had less error in distance metrics than predictions by AF2 across different distance thresholds, with average mean squared errors of 2.42 and 4.85 Å, respectively (*Figure 9b*). Furthermore, calculation of H3 contact propensities to assess

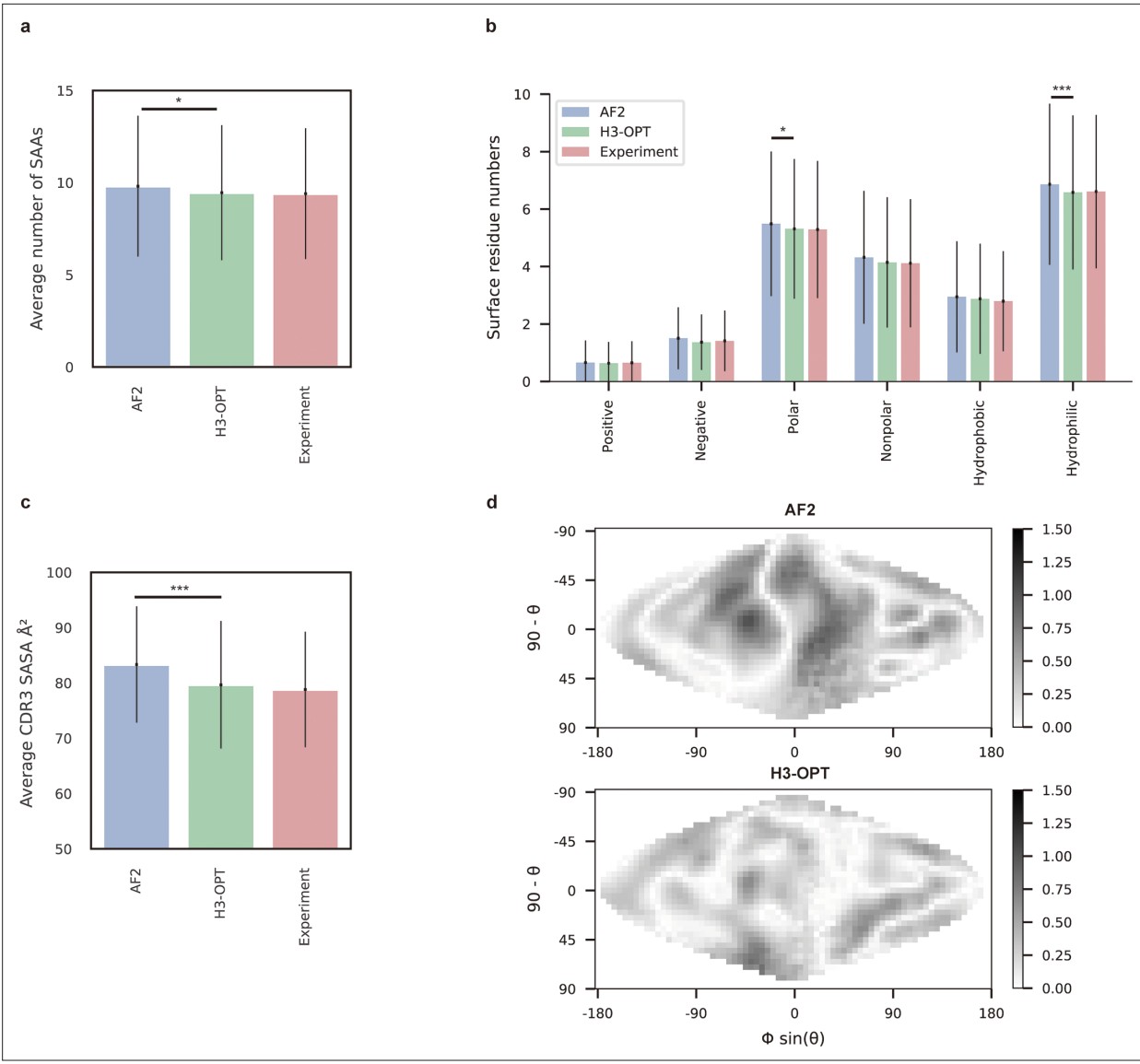

**Figure 8.** Analysis of surface patches. (**a**) Analysis of surface amino acids for predicted H3 loops. Y-axis represents average number of surface residues for H3 loops (n = 131). The surface residues of AF2 models are significantly higher than those of H3-OPT models (p<0.05). (**b**) Histogram of surface patches with different properties predicted by H3-OPT, AF2, or experimentally solved H3 loops. Error bars show standard deviations. H3-OPT models predicted lower values than AF2 models in terms of various surface properties, including polarity (p<0.05) and hydrophilicity (p<0.001). (**c**) Solvent-accessible surface area (SASA) analysis of predicted H3 loops. Values represent the difference in SASA between predicted and experimentally determined H3 structures using AF2 or H3-OPT. The SASA of AF2 models are significantly higher than those of H3-OPT models (p<0.001). (**d**) Comparison of the charged surface patches between H3-OPT and AF2 for target PDB ID: 5U3P. The surface maps compare the surface electrostatic potential of the CDR-H3 loop predicted by H3-OPT or AF2 with the native structure. Darker shading indicates greater difference in electrostatic potential. *p<0.05; **p<0.01; ***p<0.001.

The online version of this article includes the following figure supplement(s) for figure 8:

**Figure supplement 1.** Solvent-accessible surface area (SASA) analysis of predicted H3 loops.

potential differences in binding patterns between the predicted and native structures using high-quality H3 conformations indicated that H3-OPT also displayed higher accuracy in predicting contact propensities (*Figure 9c*).

Finally, to test whether H3-OPT was reliable for investigation of the detailed mechanisms underlying antigen–antibody binding, we generated contact maps and calculated binding affinities for complex structures obtained by H3-OPT, AF2, or through experiments (see 'Methods' for details). We found that contact maps predicted by H3-OPT were consistent with those observed in the

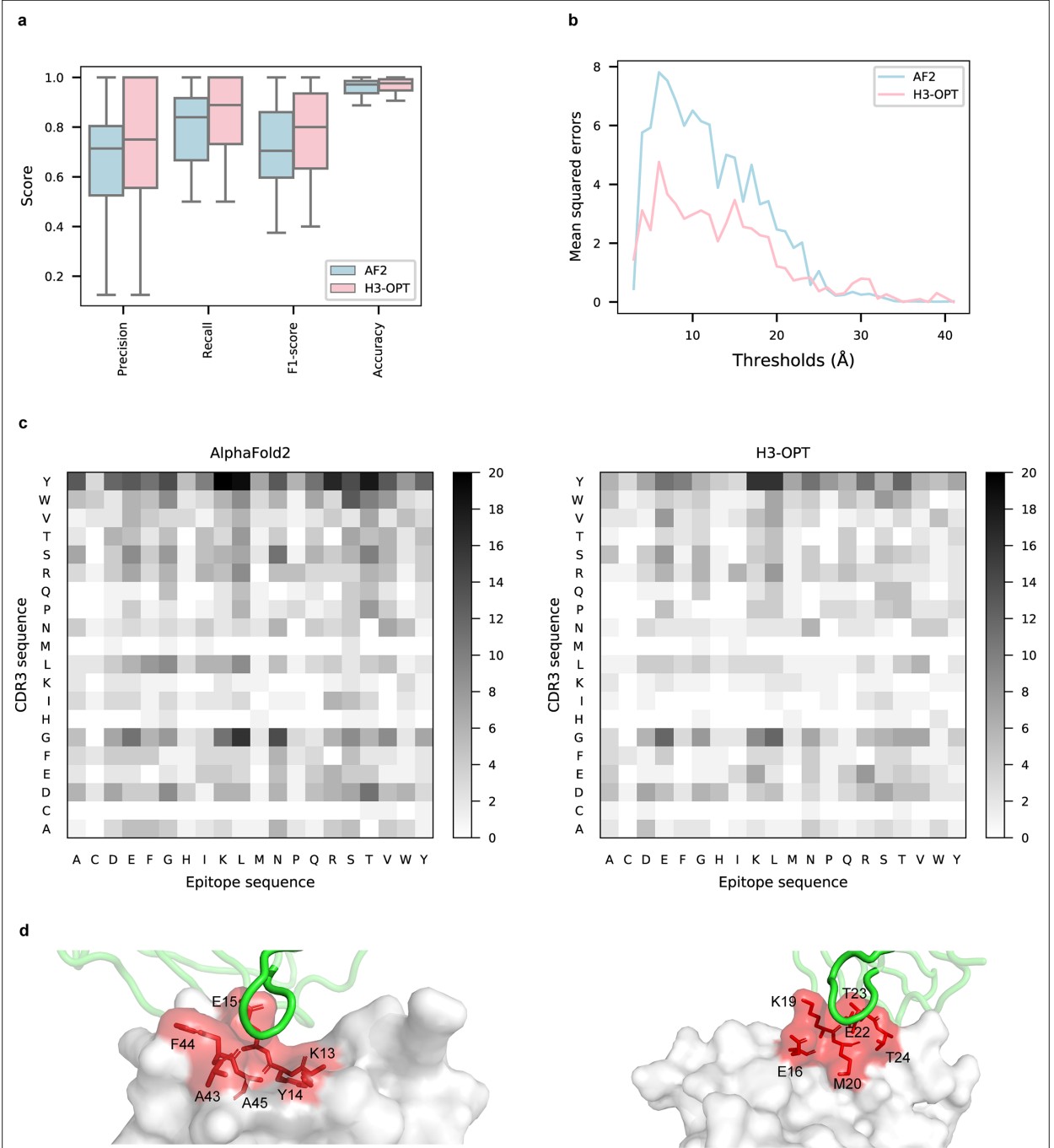

**Figure 9.** Accuracy of H3-OPT predictions of antibody–antigen interactions. (**a**) Performance of H3-OPT in binding site prediction. comparison of prediction accuracy between H3-OPT and AF2 for antibody–antigen binding sites (n = 27). Box represents interquartile range (IQR); horizontal line in the center of the box shows median. (**b**) Comparison of the mean squared errors of residue pairs between H3-OPT and AF2 under different distance thresholds. The x-axis represents the experimentally determined distance between pairs of contacting residues at the binding site in the native structure. Y-axis shows mean squared errors of H3-OPT and AF2. (**c**) Heatmaps of the frequency of pairwise residue–residue contacts across antibody–antigen interfaces. This analysis compares contact frequency of H3 loops predicted by AF2 or H3-OPT with the native structure. Darker shading indicates greater difference in contact frequency. (**d**) The predicted H3 loops of two targets interacting with antigens (PDB: 2YC1, 6O9H). The epitopes are highlighted in red and antibody chains are green. H3-OPT could identify the epitopes of different antigens that form the complementary binding interface(s) for the CDR-H3 of antibodies.

The online version of this article includes the following figure supplement(s) for figure 9:

**Figure supplement 1.** RMSD$_{backbone}$ during production runs.

**Table 6.** Comparison of binding affinities obtained from molecular dynamics (MD) simulations using AF2 and H3-OPT.

| PDB ID | AF2 (kcal/mol) | | AF2 RMSD$_{C\alpha}$ (Å) | H3-OPT (kcal/mol) | | H3-OPT RMSD$_{C\alpha}$ (Å) | AF2 (kcal/mol) | | H3-OPT (kcal/mol) | |
|---|---|---|---|---|---|---|---|---|---|---|
| | MM/GBSA | MM/PBSA | | MM/GBSA | MM/PBSA | | $\|\Delta_{MM/GBSA}*\|$ | $\|\Delta_{MM/PBSA}\|$ | $\|\Delta_{MM/GBSA}\|$ | $\|\Delta_{MM/PBSA}\|$ |
| 2ghw | −29.20 | −33.36 | 2.7 | −14.70 | −21.36 | 3.0 | 8.63 | 2.42 | 23.13 | 14.42 |
| 2yc1 | −38.85 | −37.73 | 2.3 | −43.80 | −48.72 | 1.5 | 6.80 | 18.67 | 1.85 | 7.68 |
| 3l95 | −29.59 | −53.35 | 2.5 | −47.22 | −68.86 | 2.5 | 23.60 | 11.44 | 5.97 | 4.07 |
| 3u30 | −37.31 | −42.41 | 2.6 | −44.94 | −50.07 | 2.5 | 9.64 | 2.18 | 2.01 | 5.48 |
| 4cni | −36.96 | −42.93 | 1.0 | −31.92 | −40.39 | 1.3 | 8.54 | 7.44 | 3.50 | 4.89 |
| 4nbz | −36.59 | −43.79 | 1.9 | −59.61 | −54.23 | 0.6 | 10.17 | 3.30 | 12.85 | 13.74 |
| 4xnq | −13.55 | −17.47 | 2.7 | −31.51 | −30.94 | 0.5 | 15.40 | 12.32 | 2.57 | 1.15 |
| 4ydl | −52.51 | −74.25 | 4.8 | −49.17 | −73.57 | 3.6 | 6.82 | 6.53 | 10.15 | 7.21 |
| 5e5m | −59.72 | −71.15 | 3.0 | −41.29 | −53.70 | 7.3 | 0.50 | 5.79 | 18.93 | 11.66 |
| 5f7y | −61.76 | −69.46 | 2.7 | −60.33 | −69.43 | 1.4 | 3.95 | 6.38 | 5.38 | 6.41 |
| 6kyz | −12.66 | −20.32 | 4.0 | −9.36 | −17.13 | 3.7 | 17.63 | 17.21 | 20.93 | 20.40 |
| 6o9h | −39.53 | −43.45 | 2.8 | −52.27 | −57.51 | 0.6 | 10.45 | 13.31 | 2.29 | 0.74 |
| 6pyd | −45.87 | −58.71 | 1.0 | −35.75 | −45.28 | 1.1 | 6.29 | 13.50 | 3.83 | 0.06 |
| 6u9s | −36.54 | −48.66 | 1.0 | −39.79 | −44.80 | 1.3 | 14.35 | 10.42 | 11.11 | 14.28 |
| Average | / | / | 2.6 | / | / | 2.4 | 10.20 | 9.35 | 8.89 | 8.01 |

*$\Delta_{MM/GBSA}$ (or $\Delta_{MM/PBSA}$) was calculated by subtracting the MM/GBSA (or MM/PBSA) of predicted model from the MM/GBSA· (or MM/PBSA) of experimental structure.

experimental structures (**Figure 9d**). Since affinity prediction plays a crucial role in antibody therapeutics engineering, we performed MD simulations to compare the differences in binding affinities between AF2-predicted complexes and H3-OPT-predicted complexes (**Figure 9—figure supplement 1**). Calculation of binding affinities through MD simulations showed that the average affinities of structures obtained by H3-OPT prediction were closer to those of experimentally determined structures than values obtained through AF2 (**Table 6**). These cumulative findings illustrate the informative value of high-quality H3 loops for predicting binding affinity and facilitating antibody engineering, and further support the use of H3-OPT as a powerful and versatile tool for investigating antigen–antibody interactions.

## Discussion

Several platforms for in silico antibody engineering have been developed to improve binding affinity (**Lippow et al., 2007**; **Riahi et al., 2021**; **Clark et al., 2006**; **Shan et al., 2022**), humanization (**Choi et al., 2015**; **Kurella and Gali, 2014**), and stability (**Tomar et al., 2018**; **Froning et al., 2020**). At present, engineering these biophysical properties of therapeutic antibodies heavily relies on the availability of the structures of thousands of antibodies. The accurate structure of CDR-H3 is crucial for understanding the mechanism(s) of antigen–antibody interactions. Although AF2 performs well when MSA and templates are available, its predictive capability decreases for targets with long CDR-H3 loops. By contrast, PLMs leverage tens of millions of protein sequences to learn relevant structural patterns, consequently enhancing their capacity to predict orphan antibodies (**Chowdhury et al., 2022**). The results of IgFold indicate that PLMs, which incorporate transformer-based attention mechanisms, are capable of learning both local and global sequence information (**Ruffolo et al., 2023**). While global information can be useful for challenging targets, it may lead to large errors when templates were available, as demonstrated by our case study of anti-VEGF nanobodies (**Figure 7**).

H3-OPT combines the strengths of AF2 and PLMs to predict the structures of mAbs and Nbs, and our results show that combining these features can improve the average RMSD$_{C\alpha}$ by 20% over that of AF2 or IgFold. However, H3-OPT is less efficient than PLM-based methods like IgFold because it relies on AF2 structures. To improve efficiency, one possible solution is to cluster the targets based on sequence identity and compute AF2 structures for each cluster, followed by generation of mutant structures. Despite the lower efficiency, H3-OPT offers several notable advantages over IgFold. First, H3-OPT incorporates a simplified version of Evoformer, which improves training efficiency. In addition, H3-OPT employs an alignment-based strategy to extract structural features that are then used to directly predict the Cα coordinates of H3 loops, effectively capturing the residue features relevant to loop folding. Moreover, we found that IgFold uses an augmented dataset containing AF2-predicted structures, which could potentially propagate errors during model training. In contrast, H3-OPT was trained by using high-quality and non-redundant antibody structures. Finally, H3-OPT also shows lower Cα-RMSDs compared to AF2 or tFold-Ab for the majority (six of seven) of targets in an expanded benchmark dataset, including all antibody structures from CAMEO 2022 (*Leemann et al., 2023* ; *Figure 7—figure supplement 1*).

Although H3-OPT can provide high accuracy with a small training dataset, it is likely that a larger dataset containing high-resolution structures will further improve our model. In the current study, we observed that the accuracy of AF2 CDR-H3 predictions was correlated with the length of the H3 loop. Thus, AF2 can provide accurate predictions for short loops with fewer than 10 amino acids, with PLM-based models offering little or no improvement in such cases. Conversely, antibodies with H3 loop lengths exceeding 25 residues (common in Sub3) pose a long-standing challenge for structural prediction algorithms, and none of the existing methods (including H3-OPT) tested here could provide accurate predictions due to a lack of homologous sequences and a high degree of freedom in this subgroup. Notably, H3-OPT outperformed AF2 in Sub3 because the context of protein sequences learned from ESM2 was fine-tuned to fit antibody structures. Furthermore, we envision that antibodies with H3 loop lengths ranging from 10 to 25 amino acids could be further optimized using PLMs with more model parameters. In addition, we attempted to optimize the H3 loop structure through MD simulations and QM-based methods. However, these approaches generally yielded less accurate predictions than AF2 due to inaccurately described solvent effects and other environmental factors. The development of a more accurate, physics-based method for modeling long loops may be another potentially effective strategy for improving H3-OPT.

Deep generative models are playing a vital role in the field of protein design, leading to several successful applications (*Watson et al., 2023*; *Dauparas et al., 2022*; *Shin et al., 2021*; *Luo et al., 2022*; *Watson et al., 2022*; *Ingraham et al., 2019*). These models capture the intricate relationship between sequences and structures, enabling the generation of novel protein scaffolds not existing in nature. However, the potential immunogenicity concerns associated with these newly designed proteins require comprehensive preclinical and clinical investigations. Consequently, conventional mAbs and Nbs remain the prevailing choices in medicine. Recognizing that the properties of antibodies are heavily influenced by their structures, accurate prediction of the structures of mAbs and Nbs is of great significance in optimizing their therapeutic effectiveness and clinical relevance. To this end, H3-OPT has strong potential to accelerate optimization of antibody–antigen binding as well as engineering of therapeutic antibodies with specialized biophysical properties.

## Methods
### Datasets

High-quality structures are necessary for the accurate assessment of antibody modeling methods. All antibody information was obtained from the SAbDab website. We first downloaded antibody structures with X-ray diffraction resolution of 2.5 Å or less from the RCSB website (https://www.rcsb.org/) and deleted redundant sequences with 95% or more identity. Then, we clustered the remaining sequences into 93 clusters using UCLUST (*Edgar, 2010*) by 65% sequence similarity cutoff and selected centroid sequences to ensure the representativeness and sufficient quantity of the final datasets. We additionally omitted the PDB IDs 5DHV, 3SE9, 4YDJ, 5VF6, 6CT7, 7FAB, 6NMT, 5ILT, and 5UXQ, which were unable to predict using ABodyBuilder. The Fv sequence of each target was identified from initial sequences using the *Chothia and Lesk, 1987* definitions. For the Nb dataset, we collected Nb

sequences from a non-redundant dataset (*Zavrtanik and Hadži, 2019*) and deleted all structures with low resolution (≥ 2.5 Å) to construct a high-resolution nanobody structure database. Seven native structures (PDB IDs 6NEX, 6FFJ, 4BUH, 6BHZ, 6UUM, 5O03, and 5C1M) were further excluded from the final dataset owing to the missing residues in CDR-H3 loops. In total, 125 targets were included in the final datasets: 47 mAbs (DB1) and 78 Nbs (DB2). After the model assessment, additional 98 antibodies and 42 nanobodies, which were collected from the test sets of DeepAb and NanoNet, were combined to verify the improvement of the CDR-H3 optimization methods (DB3).

## Sequence logos generation

Sequences of DB1, DB2 $V_H$ domains were determined by Chothia definitions and aligned using ANARCI (*Dunbar and Deane, 2016*) with the Aho scheme (*Honegger and Plückthun, 2001*). Then, we removed the alignment positions with 85% gaps to obtain a good visualization of alignment statistics. Finally, the alignment outputs of different datasets were submitted to the WebLogo3 server (*Crooks et al., 2004*) to obtain sequence logos of each database. These sequence logos quantitatively showed the conservation of antibody sequences. The height of each letter within the logo corresponds to its base frequency at that particular position. The letters are arranged in descending order of size, with the tallest (or most frequent) letter positioned at the top.

## Benchmarking alternative methods

To evaluate the prediction accuracy of all alternative methods, we benchmarked five AI-based methods (AF2, RoseTTAFold, trRosetta, NanoNet, and DeepAb) and two TBM methods (MODELLER and ABodyBuilder). Identical templates were excluded from TBM templates database and AF2 PDB database. To model full-length $V_H$ and $V_L$ antibody, 50 glycines were added as a linker between two chains. The output multiple sequence alignment files of AF2 were then used to predict MODELLER models. ABodyBuilder models were generated using the webserver and modeled by Sphinx. For all methods, recommended parameters were used for all antibody predictions. NanoNet was removed from the DB1 as it was only applied for Nb modeling. We only compared and summarized the target structures that were successfully produced by all participants.

## Model assessment

The residue indexes of different models were aligned and renumbered using MAFFT (*Katoh et al., 2002*) to obtain the same sequence length. We assessed the accuracy of AF2 through varying metrics, including TM-score, GDT-TS score, and GDT-HA score. The structural cutoffs for GDT-TS score were 1, 2, 4, and 8 Å, respectively, while that of the GDT-HA score were 0.5, 1, 2, and 4 Å, respectively. We calculated the average Z-scores of these methods to assess the overall similarities. The average Z-score of each target is given by

$$Z_{ave} = \frac{1}{3}Z_{GDT-TS} + \frac{1}{3}Z_{GDT-HA} + \frac{1}{3}Z_{TM-score}$$

To better evaluate the accuracy of CDR loops, RMSDs were computed after superimposing the Fv or $V_H$ backbone heavy atoms to their reference structures. All RMSDs were computed using Schrödinger v. 2017-2 (Schrödinger, New York, NY).

## Refinement of the CDR-H3 loop of AF2 through QM methods

We extracted all H3 loop atoms from different AF2 models and re-rank these models based on QM energies. The nitrogen atoms and carbonyl atoms were initially removed from N-terminals and C-terminals, respectively. Next, we added hydrogens to these loops and produced the input files for different QM optimization methods using Schrödinger v. 2017-2. The geometry optimizations were carried out with the Gaussian16 (*Stewart, 2007*) software. We first optimized the CDR-H3 loop by using PM6-D3 (*Grimme et al., 2010*; *Stephens et al., 1994*) and B3LYP/6-31G* (*Chothia and Lesk, 1987*; *Hehre et al., 1972*). The optimization method was divided into three steps: (1) optimizing hydrogens; (2) freezing the terminal C-alpha atoms and optimizing other atoms; and (3) frequency calculations to obtain Gibbs free energies. For an alternative CDR-H3 loop optimization method, we calculated the single-point energies with thermal corrections. The solvation effect can be estimated using the polarizable continuum model (with a dielectric constant $\varepsilon$ = 78.36). After optimization, the

proportion of each conformation was calculated based on the energies of five AF2 models according to Boltzmann distribution, which was used to generate a Boltzmann-averaged structure. The structures that failed in the optimization process were excluded from the final comparison.

## H3-OPT dataset

Crystal structures of antibodies were obtained from SAbDab, a non-redundant nanobody database (*Zavrtanik and Hadži, 2019*) and a subset of DeepAb. The structures with resolution >2.5 Å or redundant sequences with 95% or greater identity were removed (*Figure 4a*). We removed the structures with missing loops in CDR-H3 or lengths >45. The structures with missing residues at CDR-H3 loops were also dropped. To enhance the generalizability of H3-OPT, the sequences were clustered based on a 90% similarity cutoff by using UCLUST (*Edgar, 2010*). The resulting clusters were then randomly divided into training, validating, and testing sets to avoid overestimation of performance. The number of sequences in training, validating, and testing sets was 1021 (925 mAbs, 96 Nbs), 134 (122 mAbs, 12Nbs), and 131 (119 mAbs, 12 Nbs), respectively, with average CDR-H3 loop length of 11.8, 11.9, and 11.7, respectively. Schrödinger v. 2017-2 (Schrödinger), Numpy 1.18.5, and Pandas 1.0.5 were used for data preparation.

## Structure preparation

To address the transformational invariance for predicting 3D coordinates, we applied a structural alignment strategy. Prior to feature extraction, we removed all atoms of light chains in mAbs Fv structures and any non-standard residues in native structures. All antibody $V_H$ native structures were aligned using StructAlign module in Schrödinger with a randomly selected reference structure (PDB ID: 1GIG). Next, we superimposed AF2-predicted models to their corresponding native structures. This alignment allowed the model to learn the interactions between residues located in H3 loops while disregarding the rotation and translation of the entire structure. The heavy chain fragments were determined using Chothia definitions (*Ahdritz et al., 2022a*).

## Feature generation

As shown in *Table 7*, structural features were extracted and aggregated into the following inputs to PSPM of H3-OPT: a residue feature vector of size [$N_{res}$, 34] was constructed by concatenating 'amino acid type', 'AF2-predicted coordinates', 'AF2-predicted backbone torsion angles', 'torsion angles mask', and 'predicted residue mask'. The pairwise feature vector of size [$N_{res}$, $N_{res}$, 60] included the 'pairwise distances' and 'pairwise amino acid type'.

**Table 7.** Features of the model.
$N_{res}$ is the number of residues (*Jumper et al., 2021*).

| Feature and shape | Description |
| --- | --- |
| Amino acid type [$N_{res}$, 21] | One-hot representation of the input amino acid sequence (including 20 amino acids and unknown). |
| 3D coordinates [$N_{res}$, 3] | Cα coordinates of all AlphaFold2-predicted residues |
| Backbone torsion angles [$N_{res}$, 6] | Sine and cosine encoding of all predicted three backbone torsion angles. |
| Torsion angles mask [$N_{res}$, 3] | A mask indicating if the angle was presented in the predicted structure. |
| H3 residue mask [$N_{res}$, 1] | A mask indicating if the residue was located in H3 loop. |
| Pairwise distances [$N_{res}$, $N_{res}$, 39] | One hot representation of residue alpha carbon atoms distance. The pairwise distances ranging from 3.25 Å to 50.75 Å were put into 38 bins equally and the last bin contained any larger distances. |
| Pairwise amino acid type [$N_{res}$, $N_{res}$, 21] | One-hot representation of the input amino acid sequence. |

**Table 8.** Hyperparameters for H3-OPT models.

| Model | 2 | 5 | 1 | 3 | Best |
|---|---|---|---|---|---|
| Initial learning rate | $1^{-4}$ | $5^{-4}$ | $1^{-3}$ | $5^{-4}$ | $1^{-4}$ |
| Hidden layers | 64 | 64 | 64 | 64 | 64 |
| Iterations numbers of Evoformer-like layer | 6 | 6 | 4 | 4 | 4 |
| Average RMSD$_{C\alpha}$ (Å) | 2.42 | 2.36 | 2.35 | 2.33 | 2.24 |

RMSD = root mean square deviation.

## Network architecture

H3-OPT took the predicted H3 loops of AF2 as input and generate the antibody structure with the optimized CDR-H3 loops as output. H3-OPT was composed of a template module and a PSPM. The template module contains two submodules, that is, CBM and TGM. The CBM calculated the average confidence score of H3 residues and retained AF2 structures with confidence score >0.8. The TGM grafted the template loop structure onto the input AF2 model, when their CDR-H3 sequences were identical. We replaced the predicted CDR-H3 loop generated by AF2 with the template loop by aligning corresponding atoms, which included Cα atoms and carbonyl carbon atoms in the first two residues and Cα atoms and backbone nitrogen atoms in the last two residues.

The input residue-level features are extracted from AF2-predicted structures as mentioned above, resulting in a dimension of $L \times 34$, where $L$ is the padding length of antibody heavy chain. Similarly, the pairwise features has a dimension of $L \times L \times 60$. The PSPM network began with a row-wise gated multi-head self-attention layer that updated residue-level features from pair representations. Then, the residue-level information updated the pair representations via outer product mean module. Pair representations were updated by two multiplicative update modules. During training, the weights of the attention layer and the triangle multiplicative update modules were updated four times per epoch. Then, the final hidden states from ESM2 (esm2_t33_650M_UR50D) were passed to a linear layer and then concatenated with the residue-level features to enable the learning of latent information from vast protein sequences. To predict all Cα coordinates for the CDR-H3 loop, the output vector was passed through three linear layers with a hidden size of 64. In summary, PSPM utilized an attention-based mechanism to learn the contribution of individual residues and incorporate sequence representations from PLMs into final predictions. We used OpenFold (*Banks et al., 2005*; based on GitHub from June 2022: commit 3f57b4a041f063406059f42080ede6d495479617; *Ahdritz et al., 2022b*) to implement partial networks of AF2.

We employed a fine-tuning strategy to update the sequence representations in ESM2. Initially, we froze all weights of the 33 representation layers in ESM2 and updated only the weights of the attention layers and pair representation update modules. Subsequently, we fixed all weights of the remaining components in H3-OPT and performed fine-tuning exclusively on all hidden layers of ESM2 for the H3 loop prediction task. Finally, we fine-tuned all parameters of H3-OPT to predict the atomic coordinates of all CDR-H3 Cα atoms.

**Table 9.** Average Cα-RMSDs of our test set under different confidence cutoffs.

| Cutoff | Cα-RMSD (Å) |
|---|---|
| 0.70 | 2.46 |
| 0.75 | 2.30 |
| 0.80 | 2.24 |
| 0.85 | 2.17 |
| 0.90 | 2.29 |
| 0.95 | 2.28 |

RMSD = root mean square deviation.

During training, we trained our model on the training set and validated it on the validation set. The dropout rates of H3-OPT were set to 0.25. Mean squared error (MSE) loss was utilized to train the model. We used Adam optimizer with a learning rate of $1 \times 10^{-4}$, weight decay of $5 \times 10^{-4}$ for training. The model was trained on an NVIDIA V100 super GPU and took 1 hr per epoch over the entire training set with a batch size of 64. We randomly selected 10% of all structures for model validation. The H3-OPT was implemented using PyTorch 1.12.1 in Python 3.7.2. The hyperparameters used during training process for H3-OPT are presented in *Table 8*. The average Cα-RMSDs of our test set under different confidence cutoffs in CBM are presented in *Table 9*.

## Structure generation and refinement

We utilized a structure refinement strategy to generate structures of CDR-H3 loops and rectify structural errors. Initially, we used PSPM predictions to modify the Cα coordinates of predicted CDR-H3 loops generated by AF2. Then, we minimized the remaining atoms of the CDR-H3 loops, while keeping their Cα coordinates fixed, by using the OPLS 2005 force field (*Mirdita et al., 2022*). After the initial energy minimization step, we applied a force constant of 10 kcal/mol to these fixed atoms to guarantee complete relaxation of the entire loops.

## Validation settings

H3-OPT, AF2, HelixFold-Single, IgFold, and ESMFold were compared on the test subset. Given that tFold-Ab is only available on webserver, we compared H3-OPT with tFold-Ab on the CAMEO 2022 dataset (*Leemann et al., 2023*). We removed the target antibody structures from the AF2 database before prediction to evaluate the performance of AF2. This ensured that the predictions depended solely on the information derived from other template structures, thus avoiding biased results. Since all other methods did not require MSA searching, we did not exclude these targets from their database. To predict the entire Fv structures, we included 50 glycine residues as a linker between the heavy chain and the light chain for antibodies (*Levy, 2010*). The Cα-RMSDs of all predicted H3 loops were calculated after superimposing the $V_H$ backbone heavy atoms (without CDR-H3) to the reference structures. All structures were generated using publicly available code repositories: AF2 v2.1.1 (based on GitHub from the November 2021 version at GitHub: commit 91b43223422420d1783ed-802c8b3a8382a9309fd; *Zidek et al., 2021*), IgFold (based on GitHub from February 2023: commit 6a09298d165ed1deb438c0b6eefcbcb03ed0eca5; *Graylab, 2023*), HelixFold-Single (based on GitHub from January 2023: commit 5f39b2c2a4ecc00b89ba05b95dc56212bdd5d886; *Xiang and xiaoyao4573, 2023*), and ESMFold v1 (based on GitHub from October 2022: commit dc823b89c6acb-9f67caea53704c2a97524fbd456; *Sercu, 2022*).

## Analysis of SASAs and pairwise residue contacts

We computed the SASAs and rASAs of all CDR-H3 loops by using the ANARCI webserver (*Dunbar and Deane, 2016*) with the default settings. Residue with an rASA >25% was considered surface residues (*Schweke et al., 2022*). Subsequently, we applied AHo alignment software (*Honegger and Plückthun, 2001*) to report the averaged SASA per alignment position. The samples and alignment positions with more than 10 gaps were removed to avoid randomness and bias, reducing the final samples to 123. Finally, the difference in SASAs between H3-OPT and AF2 was calculated by subtracting predicted H3 loops SASAs from the ground truth.

To examine the contribution of H3-OPT in discovering antibody–antigen interactions, we identified contact residue pairs that were within this distance as binding sites by setting a threshold of 5 Å. We next calculated the pairwise residue distance matrices for each individual predicted complex and native structure, where each element represented the closest distance between the heavy atoms of two residues. From these pairwise distances, we derived contact propensity matrices that specifically indicated the presence or absence of interactions between residues.

## Analysis of surface electrostatic potential

We generated 2D projections of CDR-H3 loop's surface electrostatic potential using SURFMAP v2.0.0 (*Salomon-Ferrer et al., 2013*; based on GitHub from February 2023: commit: e0d51a10debc-96775468912ccd8de01e239d1900; *Chevrollier et al., 2023*) with default parameters. The 2D surface

maps were calculated by subtracting the surface projection of H3-OPT or AF2-predicted 3 loops to their native structures.

## Binding affinity calculation

We analyzed the binding affinities of antibody–antigen complexes predicted by AF2 and H3-OPT. Their relative binding affinities were calculated through MD simulations. Initially, the missing side chains and loops in the antibody structures were filled using the Protein Preparation Wizard module within Schrödinger software. Subsequently, the tLEaP module of AMBER (*Tian et al., 2020*) was employed to construct the simulation system with the ff19SB force field (*Izadi et al., 2014*) and OPC solvent model (*Chayen and Saridakis, 2008*). Additionally, the simulation system was solvated with a 0.15 M NaCl solution. Energy minimization was performed through a 5000-step steepest descent algorithm, followed by a 5-step conjugate gradient algorithm. Then, a 400-ps NVT simulation with a time step of 2 fs was performed to gradually heat the system from 0 K to 298 K (0–100 K: 100 ps; 100–298 K: 200 ps; hold 298 K: 100 ps), and a 100-ps NPT simulation with a time step of 2 fs was performed to equilibrate the density of the system. During heating and density equilibration, we constrained the antigen–antibody structure with a restraint value of 10 kcal·mol$^{-1}$·Å$^{-2}$. In the production run, 100-ns MD simulations were performed with a time step of 2 fs. The first 50 ns restrains the non-hydrogen atoms of the antigen–antibody complex, and the last 50 ns restrains the non-hydrogen atoms of the antigen, with a constraint value of 10 kcal·mol$^{-1}$·Å$^{-2}$. The distance cutoff for nonbonded interactions was set to 10 Å, and the Berendsen algorithm was utilized to maintain isotropic pressure coupling at 1 bar. The Langevin algorithm was employed to maintain the simulation temperature at 298 K. The relative binding affinities of the antigen–antibody complexes were evaluated using the MMPBSA module of AMBER software, which computed the MM/GBSA energies for the trajectory frames of last 10 ns.

## Crystallization and data collection

The protein expression, purification, and crystallization experiments were described previously (*Zhu et al., 2023*; *Yu et al., 2019*). The proteins used in the crystallization experiments were unlabeled. Upon thawing the frozen protein on ice, we performed a centrifugation step to eliminate any potential crystal nucleus and precipitants. Subsequently, we mixed the protein at a 1:1 ratio with commercial crystal condition kits using the sitting-drop vapor diffusion method facilitated by the Protein Crystallization Screening System (TTP LabTech, mosquito). After several days of optimization, single crystals were successfully cultivated at 21°C and promptly flash-frozen in liquid nitrogen. The diffraction data from various crystals were collected at the Shanghai Synchrotron Research Facility and subsequently processed using the aquarium pipeline (*Yu et al., 2019*).

## Statistics analysis

We conducted two-sided *t*-test analyses to assess the statistical significance of differences between the various groups. Statistical significance was considered when the p-values were <0.05. These statistical analyses were carried out using Python 3.10 with the Scipy library (version 1.10.1).

## Acknowledgements

This work was supported by the Tsinghua University Initiative Scientific Research Program (no. 20231080030), Vanke Special Fund for Public Health and Health Discipline Development (2022Z82WKJ009), the Tsinghua-Peking University Center for Life Sciences (no. 20111770319), Tsinghua University – Peking Union Medical College and Hospital Collaboration Foundation (no. 20191080837).

## Additional information

### Competing interests

Xiaonan Zhang, Lihang Liu: Employee of Baidu International Technology (Shenzhen) Co., Ltd. The other authors declare that no competing interests exist.

## Funding

| Funder | Grant reference number | Author |
| --- | --- | --- |
| Tsinghua University | Initiative Scientific Research Program 20231080030 | Boxue Tian |
| Vanke Special Fund for Public Health and Health Discipline Development | 2022Z82WKJ009 | Boxue Tian |
| Tsinghua-Peking University Center for Life Sciences | 20111770319 | Boxue Tian |
| Tsinghua University-Peking Union Medical College and Hospital Collaboration Foundation | 20191080837 | Feng Qian |

The funders had no role in study design, data collection and interpretation, or the decision to submit the work for publication.

## Author contributions

Hedi Chen, Conceptualization, Resources, Data curation, Software, Formal analysis, Supervision, Validation, Investigation, Visualization, Methodology, Writing - original draft, Project administration, Writing - review and editing; Xiaoyu Fan, Conceptualization, Software, Methodology; Shuqian Zhu, Yuchan Pei, Data curation, Validation; Xiaochun Zhang, Data curation, Validation, Methodology; Xiaonan Zhang, Lihang Liu, Software, Validation; Feng Qian, Conceptualization, Supervision, Funding acquisition, Project administration, Writing - review and editing; Boxue Tian, Conceptualization, Supervision, Funding acquisition, Methodology, Writing - original draft, Project administration, Writing - review and editing

## Author ORCIDs

Hedi Chen ⓘ http://orcid.org/0000-0003-3922-2156
Feng Qian ⓘ http://orcid.org/0000-0001-7415-6997
Boxue Tian ⓘ http://orcid.org/0000-0002-5830-0669

Reviewer #2 (Public Review): https://doi.org/10.7554/eLife.91512.4.sa1
Reviewer #3 (Public Review): https://doi.org/10.7554/eLife.91512.4.sa2
Author response https://doi.org/10.7554/eLife.91512.4.sa3

---

# Additional files

## Supplementary files

- MDAR checklist

## Data availability

The datasets of our study and the codes of H3-OPT are freely available at https://github.com/chdcg/H3-OPT, (copy archived at *chdcg, 2024*).

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
