## [Editor Report · eLife assessment]

This article presents H3-OPT, a deep learning method that effectively combines existing techniques for the prediction of antibody structure. This work, supported by **convincing** experiments for validation, is **important** because the method can aid in the design of antibodies, which are key tools in many research and industrial applications.

---

## [Referee Report · Reviewer #2 (Public Review)]

This work provides a new tool (H3-Opt) for the prediction of antibody and nanobody structures, based on the combination of AlphaFold2 and a pre-trained protein language model, with a focus on predicting the challenging CDR-H3 loops with enhanced accuracy than previously developed approaches. This task is of high value for the development of new therapeutic antibodies. The paper provides an external validation consisting of 131 sequences, with further analysis of the results by segregating the test sets in three subsets of varying difficulty and comparison with other available methods. Furthermore, the approach was validated by comparing three experimentally solved 3D structures of anti-VEGF nanobodies with the H3-Opt predictions

Strengths:

The experimental design to train and validate the new approach has been clearly described, including the dataset compilation and its representative sampling into training, validation and test sets, and structure preparation. The results of the in silico validation are quite convincing and support the authors' conclusions.

The datasets used to train and validate the tool and the code are made available by the authors, which ensures transparency and reproducibility, and allows future benchmarking exercises with incoming new tools.

Compared to AlphaFold2, the authors' optimization seems to produce better results for the most challenging subsets of the test set.

Weaknesses:

None

---

## [Referee Report · Reviewer #3 (Public Review)]

Summary:

The manuscript introduces a new computational framework for choosing 'the best method' according to the case for getting the best possible structural prediction for the CDR-H3 loop. The authors show their strategy improves on average the accuracy of the predictions on datasets of increasing difficulty in comparison to several state-of-the-art methods. They also show the benefits of improving the structural predictions of the CDR-H3 in the evaluation of different properties that may be relevant for drug discovery and therapeutic design.

Strengths:

The authors introduce a novel framework, which can be easily adapted and improved. Authors use a well-defined dataset to test their new method. A modest average accuracy gain is obtained in comparison to other state-of-the art methods for the same task, while avoiding testing different prediction approaches. Although the accuracy gain is mainly ascribed to easy cases, the accuracy and precision for moderate to challenging cases is comparable to the best PLM methods (see Fig. 4b and Extended Data Fig. 2), reflecting the present methodological limit in the field. The proposed method includes a confidence score for guiding users about the accuracy of the predictions.

---

## [Author Response]

The following is the authors’ response to the previous reviews.

**Reviewer #2 (Public Review):**
Weaknesses:The comparison of affinity predictions derived from AlphaFold2 and H3-opt models, based on molecular dynamics simulations, should have been discussed in depth. In some cases, there are huge differences between the estimations from H3-opt models and those from experimental structures. It seems that the authors obtained average differences of the real delta, instead of average differences of the absolute value of the delta. This can be misleading, because high negative differences might be compensated by high positive differences when computing the mean value. Moreover, it would have been good for the authors to disclose the trajectories from the MD simulations.

Thanks for your careful checks. We fully understand your concerns about the large differences when calculating affinity. To understand the source of these huge differences, we carefully analyzed the trajectories of the input structures during MD simulations. We found that the antigen-antibody complex shifted as it transited from NVT to NPT during pre-equilibrium, even when restraints are used to determine the protein structure. To address this issue, we consulted the solution provided on Amber's mailing list (http://archive.ambermd.org/202102/0298.html) and modified the top file ATOMS_MOLECULE item of the simulation system to merge the antigen-antibody complexes into one molecule. As a result, the number of SOLVENT_POINTERS was also adjusted. Finally, we performed all MD simulations and calculated affinities of all complexes.

We have corrected the “Afterwards, a 25000-step NVT simulation with a time step of 1 fs was performed to gradually heat the system from 0 K to 100 K. A 250000-step NPT simulation with a time step of 2 fs was carried out to further heat the system from 100 K to 298 K.” into “Afterwards, a 400-ps NVT simulation with a time step of 2 fs was performed to gradually heat the system from 0 K to 298 K (0–100 K: 100 ps; 100-298 K: 200 ps; hold 298 K: 100 ps), and a 100-ps NPT simulation with a time step of 2 fs was performed to equilibrate the density of the system. During heating and density equilibration, we constrained the antigen-antibody structure with a restraint value of 10 kcal×mol-1×Å-2.” and added the following sentence in the Method section of our revised manuscript: “The first 50 ns restrains the non-hydrogen atoms of the antigen-antibody complex, and the last 50 ns restrains the non-hydrogen atoms of the antigen, with a constraint value of 10 kcal×mol-1×Å-2”

In addition, we have corrected the calculation of mean deltas using absolute values and have demonstrated that the average affinities of structures predicted by H3-OPT were closer to those of experimentally determined structures than values obtained through AF2. These results have been updated in the revised manuscript. However, significant differences still exist between the estimations of H3-OPT models and those derived from experimental structures in few cases. We found that antibodies moved away from antigens both in AF2 and H3-OPT predicted complexes during simulations, resulting in RMSDbackbone (RMSD of antibody backbone) exceeding 20 Å. These deviations led to significant structural changes in the complexes and consequently resulted in notable differences in affinity calculations. Thus, we removed three samples (PDBID: 4qhu, 6flc, 6plk) from benchmark because these predicted structures moved away from the antigen structure during MD simulations, resulting in huge energy differences from the native structures.

**Author response table 1. sa3table1:** 

PDBID	AF2(kcal//mol)		AF2RMSD_(Cu)(A)	H3-OPT(kcal//mol)			AF2(kcal//mol)		H3-OPT(kcal//mol)	
	MM/GBSA	MM/PBSA		MM/GBSA	MM/PBSA	("Å")	|/_\MM//GBSA^(**)|	| /_\MM//PBSA∣	[ /_\MM//GBSA]	|/_\MM//PBSA|
2ghw	-29.20	-33.36	2.7	-14.70	-21.36	3.0	8.63	2.42	23.13	14.42
2yc1	-38.85	-37.73	2.3	-43.80	-48.72	1.5	6.80	18.67	1.85	7.68
3195	-29.59	-53.35	2.5	-47.22	-68.86	2.5	23.60	11.44	5.97	4.07
3u30	-37.31	-42.41	2.6	-44.94	-50.07	2.5	9.64	2.18	2.01	5.48
4cni	-36.96	-42.93	1.0	-31.92	-40.39	1.3	8.54	7.44	3.50	4.89
4nbz	-36.59	-43.79	1.9	-59.61	-54.23	0.6	10.17	3.30	12.85	13.74
4xx nq	-13.55	-17.47	2.7	-31.51	-30.94	0.5	15.40	12.32	2.57	1.15
4ydl	-52.51	-74.25	4.8	-49.17	-73.57	3.6	6.82	6.53	10.15	7.21
5e5m	-59.72	-71.15	3.0	-41.29	-53.70	7.3	0.50	5.79	18.93	11.66
577 y	-61.76	-69.46	2.7	-60.33	-69.43	1.4	3.95	6.38	5.38	6.41
6kyz	-12.66	-20.32	4.0	-9.36	-17.13	3.7	17.63	17.21	20.93	20.40
609h	-39.53	-43.45	2.8	-52.27	-57.51	0.6	10.45	13.31	2.29	0.74
6pyd	-45.87	-58.71	1.0	-35.75	-45.28	1.1	6.29	13.50	3.83	0.06
6u9s	-36.54	-48.66	1.0	-39.79	-44.80	1.3	14.35	10.42	11.11	14.28
Average	1	I	2.6	l	1	2.4	10.20	9.35	8.89	8.01

We also appreciate your reminder, and we have calculated all RMSDbackbone during production runs (SI Fig. 5).

**Reviewer #3 (Public Review):**
Weaknesses:The proposed method lacks of a confidence score or a warning to help guiding the users in moderate to challenging cases.We were sorry for our mistakes. We have updated our GitHub code and added following sentences to clarify how we train this confidence score module in Method Section: “Confidence score prediction module

We apply an MSE loss for confidence prediction, label error was calculated as the Cα deviation of each residue after alignment. The inputs of this module are the same as those used for H3-OPT, and it generates a confidence score ranging from 0 to 100. The dropout rates of H3-OPT were set to 0.25. The learning rate and weight decay of Adam optimizer are set to 1 × 10−5 and 1 × 10−4, respectively.”

**Reviewer #2 (Recommendations For The Authors):**
I would strongly suggest that the authors deepen their discussion on the affinity prediction based on Molecular Dynamics. In particular, why do the authors think that some structures exhibit huge differences between the predictions from the experimental structure and the predicted by H3-opt? Also, please compute the mean deltas using the absolute value and not the real value; the letter can be extremely misleading and hidden very high differences in different directions that are compensating when averaging.

I would also advice to include graphical results of the MD trajectories, at least as Supp. Material.

We gratefully thank you for your feedback and fully understand your concerns. We found the source of these huge differences and solved this problem by changing method of MD simulations. Then, we calculated all affinities and corrected the mean deltas calculation using the absolute value. The RMSDbackbone values were also measured to enable accurate affinity predictions during production runs (SI Fig. 5). There are still big differences between the estimations of H3-OPT models and those from experimental structures in some cases. We found that antibodies moved away from antigens both in AF2 and H3-OPT predicted complexes during simulations, resulting in RMSDbackbone exceeding 20 Å. These deviations led to significant structural changes in the complexes and consequently resulted in notable differences in affinity calculations. Thus, we removed three samples (PDBID: 4qhu, 6flc, 6plk) from benchmark.

Thanks again for your professional advice.

**Reviewer #3 (Recommendations For The Authors):**
(1) I am pleased with the most of the answers provided by the authors to the first review. In my humble opinion, the new manuscript has greatly improved. However, I think some answers to the reviewers are worth to be included in the main text or supporting information for the benefit of general readers. In particular, the requested statistics (i.e. p-values for Cα-RMSD values across the modeling approaches, p-values and error bars in Fig 5a and 5b, etc.) should be introduced in the manuscript.

We sincerely appreciate your advice. We have added the statistics values to Fig. 4 and Fig. 5 to our manuscript.

**Author response image 2. sa3fig2:** 

**Author response image 3. sa3fig3:** 

(2) Similarly, authors state in the answers that "we have trained a separate module to predict the confidence score of the optimized CDR-H3 loops". That sounds a great improvement to H3-OPT! However, I couldn't find any reference of that new module in the reviewed version of the manuscript, nor in the available GitHub code. That is the reason for me to hold the weakness "The proposed method lacks of a confidence score".

We were really sorry for our careless mistakes. Thank you for your reminding. We have updated our GitHub code and added following sentences to clarify how we train this confidence score module in Method Section:

“Confidence score prediction module

We apply an MSE loss for confidence prediction, label error was calculated as the Cα deviation of each residue after alignment. The inputs of this module are the same as those used for H3-OPT, and it generates a confidence score ranging from 0 to 100. The dropout rates of H3-OPT were set to 0.25. The learning rate and weight decay of Adam optimizer are set to 1 × 10−5 and 1 × 10−4, respectively.”

(3) I acknowledge all the efforts made for solving new mutant/designed nanobody structures. Judging from the solved structures, mutants Y95F and Q118N seems critical to either crystallographic or dimerization contacts stabilizing the CDR-H3 loop, hence preventing the formation of crystals. Clearly, solving a molecular structure is a challenge, hence including the following comment in the manuscript is relevant for readers to correctly asset the magnitude of the validation: "The sequence identities of the VH domain and H3 loop are 0.816 and 0.647, respectively, comparing with the best template. The CDR-H3 lengths of these nanobodies are both 17. According to our classification strategy, these nanobodies belong to Sub1. The confidence scores of these AlphaFold2 predicted loops were all higher than 0.8, and these loops were accepted as the outputs of H3-OPT by CBM."

We appreciate your kind recommendations and have revised “Although Mut1 (E45A) and Mut2 (Q14N) shared the same CDR-H3 sequences as WT, only minor variations were observed in the CDR-H3. H3-OPT generated accurate predictions with Cα-RMSDs of 1.510 Å, 1.541 Å and 1.411 Å for the WT, Mut1, and Mut2, respectively.” into “Although Mut1 (E45A) and Mut2 (Q14N) shared the same CDR-H3 sequences as WT (LengthCDR-H3 = 17), only minor variations were observed in the CDR-H3. H3-OPT generated accurate predictions with Cα-RMSDs of 1.510 Å, 1.541 Å and 1.411 Å for the WT, Mut1, and Mut2, respectively (The confidence scores of these AlphaFold2 predicted loops were all higher than 0.8, and these loops were accepted as the outputs of H3-OPT by CBM). ”. In addition, we have added following sentence in the legend of Figure 4 to ensure that readers can appropriately evaluate the significance and reliability of our validations: “The sequence identities of the VH domain and H3 loop are 0.816 and 0.647, respectively, comparing with the best template.”.

(4) As pointed out in the first review, I think the work https://doi.org/10.1021/acs.jctc.1c00341 is worth acknowledging in section "2.2 Molecular dynamics (MD) simulations could not provide accurate CDR-H3 loop conformations" of supplementary material, as it constitutes a clear reference (and probably one of the few) to the MD simulations that authors pretend to perform. Similarly, the work https://doi.org/10.3390/molecules28103991 introduces a former benchmark on AI algorithms for predicting antibody and nanobody structures that readers may find interest to contrast with the present work. Indeed, this later reference is used by authors to answer a reviewer comment.

Thanks a lot for your valuable comments. We have added these references in the proper positions in our manuscript.